# SSDD: SINGLE-STEP DIFFUSION DECODER FOR EFFICIENT IMAGE TOKENIZATION

## ABSTRACT

Tokenizers are a key component of state-of-the-art generative image models, extracting the most important features from the signal while reducing data dimension and redundancy. Most current tokenizers are based on KL-regularized variational autoencoders (KL-VAE), trained with reconstruction, perceptual and adversarial losses. Diffusion decoders have been proposed as a more principled alternative to model the distribution over images conditioned on the latent. However, matching the performance of KL-VAE still requires adversarial losses, as well as a higher decoding time due to iterative sampling. To address these limitations, we introduce a new pixel diffusion decoder architecture for improved scaling and training stability, benefiting from transformer components and GAN-free training. We use distillation to replicate the performance of the diffusion decoder in an efficient single-step decoder. This makes SSDD the first diffusion decoder optimized for single-step reconstruction trained without adversarial losses, reaching higher reconstruction quality and faster sampling than KL-VAE. In particular, SSDD improves reconstruction FID from $0.87$ to $0.50$ with $1.4\times$ higher throughput and preserve generation quality of DiTs with $3.8\times$ faster sampling. As such, SSDD can be used as a drop-in replacement for KL-VAE, and for building higher-quality and faster generative models. *Code is included in the submission, model weights will be released upon acceptance.*

## 1 INTRODUCTION

Current state-of-the-art image generation models, whether they are based on diffusion (Rombach et al., 2022), flow matching (Esser et al., 2024) or autoregressive modeling (Esser et al., 2021; Tian et al., 2024), rely on tokenizers as a key component to reduce the high-dimensional visual signal to compact latent representations. This allows for efficient training and inference of these models in the latent space, before decoding the generated latent representations back to the original signal representation in pixel-space. Beyond efficiency, tokenization has also been shown to improve generative performance (Rombach et al., 2022; Esser et al., 2021). As such, the quality and compression ratio of the tokenizer directly affect the generative quality of latent models. In this work we focus on *continuous* tokenization, as used in state-of-the-art diffusion and flow-matching models.

For generative modeling, tokenizers are usually trained as autoencoders. They use an encoder $E : x \mapsto z$ to compress an image $x \in \mathbb{R}^{3 \times H \times W}$ into a latent representation $z \in \mathbb{R}^{c \times \frac{H}{f} \times \frac{W}{f}}$, where $f$ is the spatial downsampling factor and $c$ is the channel dimension of the latent. The decoder $D : z \mapsto \hat{x}$ recovers an estimate $\hat{x}$ of the original signal $x$. As the encoding process is lossy (when using typical configurations with $f^2 \geq 4c$), the decoding process can be seen as a generative task conditioned on the latent representation $z = E(x)$. The quality of reconstruction over a data distribution is quantified by two aspects. (i) **Distortion:** expressed as $\mathbb{E}_x \Delta(x, \hat{x})$, where $\hat{x} = D(E(x))$, and $\Delta$ is a measure of the distortion between *two images*. Distortion can be quantified by low-level similarity metrics such as MSE, MAE, PSNR and SSIM, or by high-level metrics based on features found in trained deep neural networks, such as LPIPS (Zhang et al., 2018) and DreamSim (Fu et al., 2023). (ii) **Distribution shift:** expressed as $d(P_x, P_{\hat{x}})$, with $d$ being a measure of the divergence between *reference* and *reconstructed distributions*. Metrics commonly used to evaluate distribution shift of generative models include Fréchet Inception Distance (FID) (Heusel et al., 2017), or Density and Coverage (Naeem et al., 2020). As shown by Blau & Michaeli (2018), there exists a fundamental trade-off between these two different notions of reconstruction quality (referred to as *Distortion*

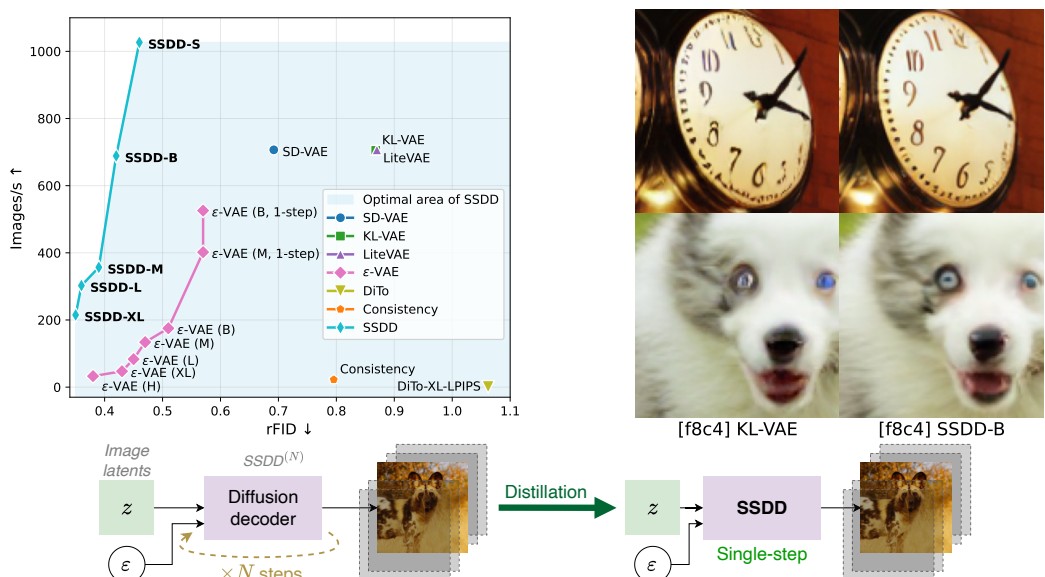

Figure 1: **Left:** Speed-quality Pareto-front for different state-of-the-art f8c4 feedforward and diffusion autoencoders. **Right:** Zoomed-in patches from reconstructions of KL-VAE and SSDD models with similar throughput. **Bottom:** High-level overview of our method.

and *Perception* by them) when the signal is compressed sufficiently. While the first term can be minimized by a deterministic decoder, the second term is best optimized using a generative decoder. We refer to Appendix A.1 for additional details and a synthetic motivational example.

Common tokenizers such as the KL-VAE from Rombach et al. (2022) are trained using an L1 reconstruction loss, LPIPS (Zhang et al., 2018), and a GAN discriminator (Goodfellow et al., 2014), with an added KL-regularization on the latent space. L1 and LPIPS losses directly optimize for distortion, while the GAN loss aims to decrease distribution shift. However, the diversity of these deterministic decoders, which affects the distribution shift, is limited by the expressivity of the latent $z$. Additionally, stable scaling of GAN losses is non-trivial (Brock et al., 2019). Diffusion auto-encoders (Chen et al., 2025d; Zhao et al., 2025a) have recently been proposed as an alternative, that directly model the conditional data distribution and optimizing for distributional shift. Diffusion decoders naturally model the distribution of missing signal from the latents (Birodkar et al., 2024), allowing higher compression ratios and focusing the task of latent modeling on higher-level features. While these models have shown improved reconstruction and generation results at a given model size, they still suffer from (i) slow sampling, due to the iterative denoising process and (ii) the reliance on a GAN loss to achieve state-of-the-art results, despite the diffusion objective.

In this work, we introduce SSDD, a new diffusion autoencoder that overcomes previous limitations. Our contributions can be summarized as: (1) An efficient and scalable alternative to the simple convolutional U-Net architectures for diffusion decoders, based on the U-ViT (Hoogeboom et al., 2023b; 2025) and achieving superior results at all model scales. (2) An analysis of key design choices to train our model, which is the first diffusion decoder to reach state-of-the-art reconstruction results without adversarial training, improving the scaling and stability of the training. (3) A single-step distillation method for SSDD with minimal quality impact, making it both the highest quality and fastest diffusion decoder, surpassing KL-VAE and previous diffusion auto-encoders on quality and reconstruction speed at the same downsampling factor. Together these contributions improve reconstruction FID from $0.87$ to $0.50$ with $1.4\times$ higher throughput, and increase sampling speed by $3.8\times$ while preserving output quality when used to decode from DiT models.

## 2 RELATED WORK

**Image tokenizers.** Tokenization is the first stage in state-of-the-art generative image models. Tokenizers for diffusion models compress images into continuous low-dimension representations (Rom-

bach et al., 2022; Peebles & Xie, 2023), using a convolutional autoencoder (Hinton & Salakhutdinov, 2006) trained with L1, KL, LPIPS (Zhang et al., 2018) and GAN (Goodfellow et al., 2014) losses, which we refer to as KL-VAE. Discrete autoencoders are used as tokenizers for autoregressive modeling (van den Oord et al., 2017; Razavi et al., 2019; Chang et al., 2022; Yu et al., 2023; 2022; Esser et al., 2021; Sun et al., 2024), replacing the KL regularization with quantization. Advances have been made on multiple aspects of those models, including efficiency (Sadat et al., 2024), semantic alignment (Yao et al., 2025), architecture and training (Chen et al., 2025a; Yang et al., 2025), spatial downsampling (Chen et al., 2025c), quantization methods (Yu et al., 2024a; Zhao et al., 2025b; Mentzer et al., 2024; Lee et al., 2022; Huh et al., 2023), and 1D-tokenization (Yu et al., 2024c; Chen et al., 2025b). However, most tokenizers rely on deterministic decoders optimized for image-to-image distortion, but may be suboptimal in terms of distribution shift (Blau & Michaeli, 2018).

**Diffusion decoders.** Diffusion decoders (Preechakul et al., 2022; Shi et al., 2022; Hudson et al., 2024; Liu et al., 2025; Pandey et al., 2022), usually based on U-Net models (Ronneberger et al., 2015), are trained using a diffusion loss. They have been applied in a compression setting (Hoogeboom et al., 2023a; Birodkar et al., 2024; Yang & Mandt, 2023; Careil et al., 2024) in place or in addition to deterministic decoders to improve quality by modeling the distribution of realistic reconstructions. Some variants also use diffusion decoders as an additional stage within a KL-VAE (Pernias et al., 2024; Bachmann et al., 2025), further compressing the encoding space. Recent work has shown that these models are scalable and easy to train when only using the diffusion loss (Chen et al., 2025d), and, at the cost of adding additional LPIPS and GAN losses, can be competitive to similarly-sized VAEs (Zhao et al., 2025a; Sargent et al., 2025; Betker et al., 2023). Diffusion decoders still suffer from slow multi-step sampling and the reliance on hard-to-scale (Brock et al., 2019) adversarial training. Our work aims to alleviate these limitations.

**Diffusion and flow image modeling.** Diffusion and flow matching models estimate the reverse process of a forward process that interpolates between data and pure Gaussian noise (Song & Ermon, 2019; Ho et al., 2020; Song et al., 2021). They were shown by Dhariwal & Nichol (2021) to outperform previous methods such as GANs (Goodfellow et al., 2014) and VAEs (Kingma & Welling, 2014). Diffusion models, often operating in a compressed latent space for efficiency (Rombach et al., 2022), underlie current state-of-the-art models for text-to-image (Baldridge et al., 2024; Nichol et al., 2022; Rombach et al., 2022; Podell et al., 2024), image-to-image (Saharia et al., 2022), and video generation (Gupta et al., 2024; Girdhar et al., 2024; Polyak et al., 2024; Agarwal et al., 2025). Ongoing research efforts aim to improve various aspects of these models, including training, sampling speed and model architecture (Karras et al., 2022; Peebles & Xie, 2023; Kingma & Gao, 2023; Nichol & Dhariwal, 2021; Chen, 2023). In particular, flow matching (Lipman et al., 2023; Liu et al., 2023; Albergo & Vanden-Eijnden, 2023) reformulates the diffusion as a velocity field estimation between noise and data, yielding improvements and faster sampling in generative image models (Esser et al., 2024). Pixel-space diffusion has seen progress with new training architectural improvements (Hoogeboom et al., 2023b; 2025; Crowson et al., 2024). As an alternative to diffusion, autoregressive models have seen fast development both for discrete and continuous modeling (van den Oord et al., 2017; Chang et al., 2022; Esser et al., 2021; Tian et al., 2024; Yu et al., 2024b; Team, 2025; Liang et al., 2025). Recent approaches have also been mixing both methods to improve generation (Chen et al., 2024; Li et al., 2024; Zhou et al., 2025; Tang et al., 2025). Our work leverages continuous flow matching for generative modeling in pixel space.

**Few-step inference.** The main drawback of diffusion models is the iterative inference process which requires multiple, often several tens or hundreds, forward passes through the model. To address this, few- or single-step sampling methods have been developed, through distillation or trajectory alignment. Consistency models (Song et al., 2023; Song & Dhariwal, 2024; Luo et al., 2023) distill pre-trained diffusion model capabilities into new few-step tailored generator models. Other methods fine-tune the diffusion model using a teacher-student setup for fast sampling (Yin et al., 2024b;a; Liu et al., 2024; Luhman & Luhman, 2021; Meng et al., 2023; Salimans & Ho, 2022; Zhao et al., 2023). We leverage this approach and study its impact on decoding performance of our models.

## 3 SINGLE-STEP DIFFUSION DECODER METHOD

We introduce here the main components of our method SSDD combining several key innovations to make diffusion decoders both efficient and practical. First, we introduce a hybrid U-Net–transformer

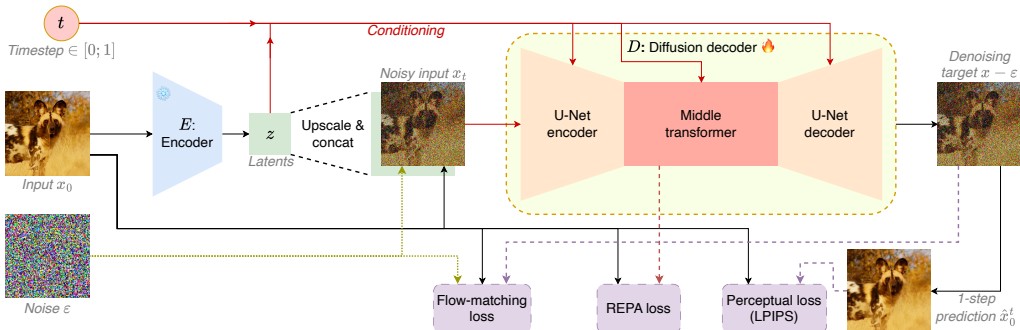

Figure 2: **Training of SSDD tokenizer.** Input image $x_0$ is mapped to latents $z$ by the (possibly frozen) encoder $E$. Noise $\epsilon \sim \mathcal{N}(0,1)$ is sampled and added to $x_0$ to form the noisy input $x_t$. The decoder $D$ learns to denoise $x_t$ conditioned on $z$ (input + AdaNorm) and $t$ (AdaNorm). The model is trained with flow-matching (generative), REPA (features alignment) and LPIPS (perceptual) losses.

architecture that leverages convolutional inductive biases while scaling effectively with transformer blocks (Section 3.1). Second, we propose a GAN-free training scheme based on flow matching, perceptual alignment, and feature regularization, ensuring stable training without adversarial losses (Section 3.2). Third, we design a single-step distillation process that transfers the behavior of multi-step diffusion to a fast one-step decoder, achieving both quality and speed (Section 3.3). Finally, we show that SSDD can operate with shared encoders, enabling compatibility with existing VAEs and reducing training costs (Section 3.4). Together, these components make SSDD the first diffusion decoder optimized for single-step reconstruction, without compromising generative performance.

## 3.1 A SCALABLE PIXEL-SPACE DIFFUSION DECODER ARCHITECTURE

Existing pixel-space diffusion decoders use the convolutional U-Net architecture (Zhao et al., 2025a) that was found to be successful in early pixel-space diffusion models (Dhariwal & Nichol, 2021). Transformer architectures, however, have proven to be more effective and scalable for latent modeling (Peebles & Xie, 2023). We aim to combine the strengths of convolutional U-Nets to model features in pixel-space with the generative capabilities of diffusion transformers. We base our architecture on the U-ViT (Hoogeboom et al., 2023b) with several modifications.

**U-Net with latent transformer.** Based on the U-ViT architecture, we modify a U-Net (Dhariwal & Nichol, 2021) using four levels of two convolutional ResNet blocks and three convolution downsampling / upsampling layers by removing all attention layers and replacing the middle attention block by a transformer operating on tokens representing $8 \times 8$ pixel patches. Each transformer block uses the GEGLU activation function (Shazeer, 2020), a $4\times$ multiplicative factor for the hidden MLP dimension, and multi-head attention (Vaswani et al., 2017).

**Time and position embedding.** We embed time using adaptive group-normalization (AdaGN) (Dhariwal & Nichol, 2021) as the second normalization layer in ResNet blocks. To enforce a local inductive bias, we learn a relative positional embedding table (Shaw et al., 2018) for each self-attention layer, with visual tokens only attending to tokens up to a distance of $8$ on the width and height axes. This results in a fixed $17 \times 17$ attention window and relative positional embedding table.

**Conditioning.** Following Chen et al. (2025d); Zhao et al. (2025a), we condition our model by upsampling the feature grid $z$ from $(c, \frac{H}{f}, \frac{W}{f})$ to $(c, H, W)$, and concatenating it with the noised image along the channel dimension in our model. As our model is relatively deep and narrow compared to prior work, we improve conditioning by adding a second mechanism using adaptive normalization. We replace the first GroupNorm of ResNet blocks with AdaGN, and the first LayerNorm of transformer blocks by AdaLN (Peebles & Xie, 2023).

**Scaling.** To study a wide range of decoding capacities, we scale our diffusion decoders from 13.4M (SSDD-S) to 345.9M parameters (SSDD-H) by increasing the number of channels and of transformer layers. For most of our experiments we consider SSDD-M (48M parameters), which has a similar size to the convolutional decoder from Rombach et al. (2022) (47.2M parameters). More details about model parametrization can be found in Appendix B.

## 3.2 GAN-FREE TRAINING

**Flow matching.** We train our model using the optimal transport flow-matching loss (Lipman et al., 2023) with $\sigma_{min} = 0$. Specifically, we use $x_t = (1 - t)x + t\varepsilon$, and our decoder $D(x_t|t, z)$ is trained to predict the velocity field $\nu = x - \varepsilon$ with the L2 loss. This defines our main training loss $\mathcal{L}_{\text{FM}} = \parallel x - \varepsilon - D(x_t|t, z) \parallel^2$. Following Esser et al. (2024), we sample $t$ during training using the logit-normal distribution (Atchison & Shen, 1980) with location $m = 0$ and scale $s = 1$.

**Perceptual and regularization losses.** As noted by Chen et al. (2025d), perceptual loss is important to guide the generation toward perceptually correct areas. We therefore add the LPIPS (Zhang et al., 2018) loss $\mathcal{L}_{\text{LPIPS}} = \text{LPIPS}(x, \hat{x}_0)$, with $\hat{x}_0 = x_t + tD(x_t|t, z)$ the single-step prediction of $x_0$ from $x_t$ and $z$. As the middle part of our model acts as an implicit latent-space diffusion transformer, we further stabilize and accelerate our training by adding the REPA loss (Yu et al., 2025), denoted as $\mathcal{L}_{\text{REPA}}$. We use DINOv2-B (Oquab et al., 2024) reference features for REPA, which are aligned with the tokens extracted from the 4th layer of our transformer, through a 2-layer MLP. The FM loss trains the decoder to predict the clean image trajectory, while LPIPS aligns perceptual features and REPA stabilizes transformer representations. We combine them in our final loss as $\mathcal{L} = \mathcal{L}_{\text{FM}} + \lambda_{\text{LPIPS}}\mathcal{L}_{\text{LPIPS}} + \lambda_{\text{REPA}}\mathcal{L}_{\text{REPA}}$, with $\lambda_{\text{LPIPS}} = 0.5$ and $\lambda_{\text{REPA}} = 0.25$.

**Multi-scale adaptation.** While autoencoders have been observed to generalize to higher resolutions than the one used for training (Sadat et al., 2024), best performance is often reached when training or fine-tuning at specific higher target resolutions. To alleviate the need of training from scratch at multiple resolutions and reduce the required training compute, we use a two-stage training method, based on the procedure from Sadat et al. (2024). During the first stage, we train a model on $128 \times 128$ crops. We additionally reduce the divergence between the features distribution of the training and fine-tuning images by adding random resizing to the data augmentation, from which we extract a $128 \times 128$ crop. During the second and third stages, we fine-tune and distill the first-stage weights at the target resolution (either $128 \times 128$ or $256 \times 256$). Given the increased cost of training at higher resolutions, we use the second model (optimized for $256 \times 256$) to evaluate on larger images (e.g. $512 \times 512$ or $1024 \times 1024$), which we observe to generalize sufficiently well.

## 3.3 SINGLE-STEP DISTILLATION

A key limitation of diffusion decoders is their iterative nature, which limits throughput in downstream generative models. We propose to align the behavior of a multi-step diffusion decoder with the one of a single-step generator, without sacrificing quality.

**Sampling behavior.** We first analyze the effect of the number of sampling steps $N$. As previously observed by Zhao et al. (2025a), reconstruction quality is non-monotonic: one-step denoising yields the best low-level distortion metrics (e.g., PSNR), while perceptual metrics (such as LPIPS and rFID) improve up to an intermediate number of steps before degrading. Choosing a schedule therefore amounts to selecting a point on the perception–distortion trade-off curve. We discuss this behavior further in Appendix A.2 and Appendix C. As we target diversity and quality for generative applications over reconstruction fidelity, we start with large denoising steps before adding a few denoising steps to refine details. We adopt the scheduler from Sargent et al. (2025), yielding the timesteps schedule $t_i = (\frac{N-i+1}{N})^\rho$ for $i \in [1; N]$. We use a shift parameter of $\rho = 2$ with $N = 8$ sampling steps for all our SSDD models, to balance realism (*low distributional shift*) and fidelity (*low distortion*).

**One-step sampling behavior alignment by distillation.** To address the key limitations of diffusion decoders, we align the behavior of a single-step generation with that of multi-step through a lightweight distillation strategy. Specifically, we use a frozen copy of our decoder $D_{\text{ref}}$ as a teacher to produce multi-step reconstructions $\hat{x}_{\text{ref}}$, and fine-tune the student decoder $D$ to reproduce this behavior in a single step. Unlike Luhman & Luhman (2021) that solely rely on an L2 regression loss, we preserve the full training objective during distillation, with flow-matching and LPIPS terms being computed against teacher outputs. Each iteration, we sample $z \sim E(x)$, $\varepsilon \sim \mathcal{N}(0; I)$, $\hat{x}_{\text{ref}} = \text{Sample}_{\text{steps}=N}(D_{\text{ref}}, \varepsilon, z)$. We fine-tune $D$ using $t = 1$, the same $z, \varepsilon$ values and the altered loss $\mathcal{L}^{\text{distill}}$, with $\mathcal{L}_{\text{FM}}^{\text{distill}} = \parallel \hat{x}_{\text{ref}} - \varepsilon - D(x_t|t, z) \parallel^2$ and $\mathcal{L}_{\text{LPIPS}}^{\text{distill}} = \text{LPIPS}(x, \hat{x}_0)$. This allows the distilled decoder to mimic the perceptual and generative properties of the teacher, while operating in a single step and thus achieving much higher efficiency. To our knowledge, this is the first work

showing that the behavior of multi-step diffusion decoders can be aligned to a single-step generator without a strong quality drop, making SSDD directly usable in generative pipelines.

## 3.4 SHARED ENCODERS

As our focus is on the decoder design, we use the standard KL-regularized convolutional design from Rombach et al. (2022) for the encoder. We refer to encoders by their patch size $f$ (controlling spatial downsampling), and output channel dimension $c$. Unless specified otherwise, we use the f8c4 encoder (compressing patches of size $8 \times 8$ into $4$-channels) in our experiments.

**Shared encoder.** Previous works exploring diffusion autoencoders train a specific encoder with each decoder. While this allows to reach optimal results, it creates a distinct latent space associated with each encoder. In particular, this requires training a separate generative diffusion model associated with each model. Despite the encoder performance being dependent on the image resolution, we find that we can train a near-optimal multi-resolution encoder with a simple procedure. We train an encoder together with a SSDD-M decoder, benefiting from the multi-scale data augmentation, learning to encode features at different resolutions. We then freeze this encoder and use it to train decoders with a different number of parameters. Unless specified otherwise, our decoders all rely on a single shared encoder for each combination of downsampling factor ($f$) and channel count ($c$).

**Pretrained encoders.** SSDD can also be directly trained to decode images from any existing encoder mapping an input image to a grid-shaped latent representation. We experiment using encoders both from KL-VAE (Rombach et al., 2022) and DiTo (Chen et al., 2025d) autoencoders.

# 4 EXPERIMENTAL VALIDATION

## 4.1 EXPERIMENTAL SETUP

We train and evaluate SSDD models of various sizes from SSDD-S to SSDD-H on ImageNet-1k (Deng et al., 2009). For details on model parametrization and training, see Appendix B. We use $\text{SSDD}^{(N)}$ to refer to models using $N$ sampling steps, and plain SSDD for single-step distilled models.

Table 1: **Comparison with state-of-the-art models on ImageNet $256 \times 256$.** Using a downsampling factor $f$ of 8 or 16, and $c = 4$ or $c = 16$ latent channels. For each metric, we highlight the **first** and second best results. *#D: decoder parameter count.* †: *trained on larger dataset.*

| | Method | #D | N | Images/s | rFID↓ | LPIPS↓ | DreamSim↓ | PSNR↑ | SSIM↑ |
|---|---|---|---|---|---|---|---|---|---|
| **f8c4** | SD-VAE† | 47.2M | 1 | 707 | 0.69 | 0.061 | 0.042 | 25.52 | 0.78 |
| | KL-VAE (Rombach et al., 2022) | 47.2M | 1 | 705 | 0.87 | 0.065 | 0.046 | 24.11 | 0.75 |
| | LiteVAE (Rombach et al., 2022) | 53.33M | 1 | 707 | 0.87 | - | - | 26.02 | 0.74 |
| | $\varepsilon$-VAE (B, 1-step) (Zhao et al., 2025a) | 20.63M | 1 | 526 | 0.57 | - | - | - | - |
| | $\varepsilon$-VAE (M) (Zhao et al., 2025a) | 49.33M | 3 | 134 | 0.47 | - | - | 27.65 | 0.84 |
| | $\varepsilon$-VAE (H) (Zhao et al., 2025a) | 355.62M | 3 | 33 | 0.38 | - | - | **29.49** | **0.85** |
| | DiTo-XL-LPIPS (Chen et al., 2025d) | 592.2M | 50 | 1 | 1.062 | 0.077 | 0.055 | 23.53 | 0.72 |
| | Consistency decoder (Betker et al., 2023) | 625.1M | 2 | 23 | 0.80 | 0.065 | 0.044 | 23.71 | 0.73 |
| | SSDD-S | 13.4M | 1 | **1027** | 0.46 | 0.060 | 0.039 | 24.08 | 0.74 |
| | SSDD-B | 20.2M | 1 | 689 | 0.42 | 0.058 | 0.036 | 24.18 | 0.75 |
| | SSDD-M | 48.0M | 1 | 357 | 0.39 | 0.055 | 0.034 | 24.38 | 0.75 |
| | SSDD-L | 85.2M | 1 | 302 | 0.36 | 0.053 | 0.032 | 24.49 | 0.76 |
| | SSDD-XL | 153.8M | 1 | 215 | **0.35** | **0.052** | **0.031** | 24.60 | 0.76 |
| **f16c16** | KL-VAE (Rombach et al., 2022) | 36.4M | 1 | **1196** | 0.82 | 0.066 | 0.048 | 23.88 | **0.74** |
| | VA-VAE (Yao et al., 2025) | 39.5M | 1 | 1178 | 0.55 | 0.132 | - | **26.10** | 0.72 |
| | SSDD-S | 13.4M | 1 | 985 | 0.45 | 0.061 | 0.038 | 23.75 | 0.73 |
| | SSDD-B | 20.2M | 1 | 686 | 0.41 | 0.059 | 0.035 | 23.86 | 0.73 |
| | SSDD-M | 48.0M | 1 | 368 | 0.40 | 0.056 | 0.033 | 24.08 | **0.74** |
| | SSDD-L | 85.2M | 1 | 315 | **0.34** | **0.053** | **0.030** | 24.20 | **0.74** |
| **f16c4** | KL-VAE (Rombach et al., 2022) | 36.4M | 1 | **1176** | 2.93 | - | - | 20.57 | 0.66 |
| | $\varepsilon$-VAE (M) (Zhao et al., 2025a) | 49.33M | 3 | 134 | 1.91 | - | - | 21.27 | 0.69 |
| | $\varepsilon$-VAE (H) (Zhao et al., 2025a) | 355.62M | 3 | 33 | 1.35 | - | - | **22.60** | **0.71** |
| | SSDD-S | 13.4M | 1 | 951 | 2.29 | 0.186 | 0.121 | 15.59 | 0.45 |
| | SSDD-B | 20.2M | 1 | 663 | 1.77 | 0.177 | 0.111 | 16.06 | 0.46 |
| | SSDD-M | 48.0M | 1 | 360 | 1.23 | 0.161 | 0.096 | 17.13 | 0.49 |
| | SSDD-L | 85.2M | 1 | 305 | **0.96** | **0.155** | **0.089** | 17.26 | 0.50 |

**Evaluation.** We evaluate the quality of our models on the 50k images of the ImageNet evaluation set. For reconstruction, we assess the *distribution shift* using the reconstruction Fréchet Inception Distance (rFID) (Heusel et al., 2017). We additionally evaluate the following distortion metrics: PSNR and SSIM (Wang et al., 2004) for low-level similarity, and LPIPS (Zhang et al., 2018) and DreamSim (Fu et al., 2023) as high-level perceptual metrics. To evaluate the impact of our model on generation quality, we train Diffusion Transformer models (Peebles & Xie, 2023) on ImageNet-1k at $256 \times 256$ resolution using the original model configuration. Following Peebles & Xie (2023), we train DiT-XL/2 models for 400k steps and evaluating the generation FID (gFID) without classifier-free guidance (CFG) (Ho & Salimans, 2021) or, if specified, with a CFG of 1.375.

**Baselines.** We evaluate our model against the standard KL-VAE from Rombach et al. (2022) trained on ImageNet, and the commonly-used stronger performing ft-EMA VAE (Peebles & Xie, 2023), which we refer to as SD-VAE. We also include the more recent LiteVAE (Sadat et al., 2024) and VA-VAE (Yao et al., 2025), and compare against existing diffusion decoders DITO (Chen et al., 2025d) and $\varepsilon$-VAE (Zhao et al., 2025a). We evaluate the publicly-available weights of SD-VAE (f8c4) and the KL-VAE (f8c4, f16c16, f32c64), and reproduce DITO training from their codebase. For other baselines, we use available published results by lack of public code or weights.

## 4.2 IMAGE RECONSTRUCTION AND GENERATION

**Fast image reconstruction.** We compare reconstruction quality of our SSDD models against other baselines in Table 1 using three encoder configurations. All methods use the same encoder architecture, and we use single-step distilled models for SSDD. Our method outperforms all existing in terms of rFID, LPIPS and DreamSim, across all encoder configurations, using smaller models and a single sampling step. We also benefit from a significant throughput increase compared to multi-step models, while maintaining high reconstruction quality. In particular, for the **f8c4** encoder configuration, our SSDD-S model **outperforms all single-step decoding methods both in speed and perceptual quality**, including single-step sampling from $\varepsilon$-VAE. Scaling the number of parameters to 345.9M, our SSDD-H obtains similar or better rFID compared to $\varepsilon$-VAE-H, while benefiting from a $3\times$ increase in decoding speed. In Figure 1 we compare deterministic and generative reconstruction methods in a speed-rFID plot, showing that SSDD is Pareto optimal across the board.

Switching to **f16c16** and **f16c4** encoders, the standard KL-VAE decoders are deeper and narrower, reducing their parameter count with an associated drop in quality. Since our models keep similar size and speed across the different encoder configurations, SSDD consistently outperforms existing baselines with a higher compression ratio (**f16c4**), while maintaining similarly high throughput. Using these deeper encoders, the gap of performance between deterministic and diffusion decoders widens. This reveals the impact of generative-focused reconstruction for heavily compressed representations. We also observe an increased gap in sample quality between smaller and larger models, associated with their respective modeling capacity. Larger models bring larger quality gains, as

Table 2: **Scaling of models and image resolution on ImageNet.** All models use an f8c4 encoder. SSDD instances use the same shared encoder and are distilled into single-step decoders. Decoder with similar parameters count are shown highlighted with the same color. Evaluations at $512 \times 512$ or higher resolution are conducted using the same model as for $256 \times 256$.

| Method | #D | $128 \times 128$ rFID↓ | $128 \times 128$ LPIPS↓ | $256 \times 256$ rFID↓ | $256 \times 256$ LPIPS↓ | $512 \times 512$ rFID↓ | $512 \times 512$ LPIPS↓ | $1024 \times 1024$ rFID↓ | $1024 \times 1024$ LPIPS↓ |
|---|---|---|---|---|---|---|---|---|---|
| KL-VAE | 47.2M | - | - | 0.87 | 0.065 | 0.29 | 0.064 | 0.20 | 0.059 |
| $\varepsilon$-VAE (B) | 20.63M | 1.94 | - | 0.52 | - | 0.61 | - | - | - |
| $\varepsilon$-VAE (M) | 49.33M | 1.58 | - | 0.47 | - | 0.53 | - | - | - |
| $\varepsilon$-VAE (L) | 88.98M | 1.47 | - | 0.45 | - | 0.41 | - | - | - |
| $\varepsilon$-VAE (XL) | 140.63M | 1.34 | - | 0.43 | - | 0.39 | - | - | - |
| $\varepsilon$-VAE (H) | 355.62M | 1 | - | 0.38 | - | 0.35 | - | - | - |
| DITO-XL + LPIPS | 592.2M | - | - | 1.06 | 0.077 | 0.28 | 0.080 | 0.15 | 0.086 |
| Consistency decoder | 625.1M | - | - | 0.80 | 0.065 | 0.30 | 0.066 | 0.15 | 0.061 |
| SSDD-S | 13.4M | 1.89 | 0.061 | 0.46 | 0.060 | 0.28 | 0.062 | 0.16 | 0.064 |
| SSDD-B | 20.2M | 1.48 | 0.059 | 0.42 | 0.058 | 0.22 | 0.060 | 0.13 | 0.058 |
| SSDD-M | 48.0M | 1.04 | 0.056 | 0.39 | 0.055 | _0.20_ | 0.057 | _0.12_ | 0.055 |
| SSDD-L | 85.2M | _0.88_ | _0.054_ | _0.36_ | _0.053_ | **0.19** | _0.055_ | **0.11** | _0.053_ |
| SSDD-XL | 153.8M | **0.81** | **0.052** | **0.35** | **0.052** | _0.20_ | **0.053** | **0.11** | **0.051** |

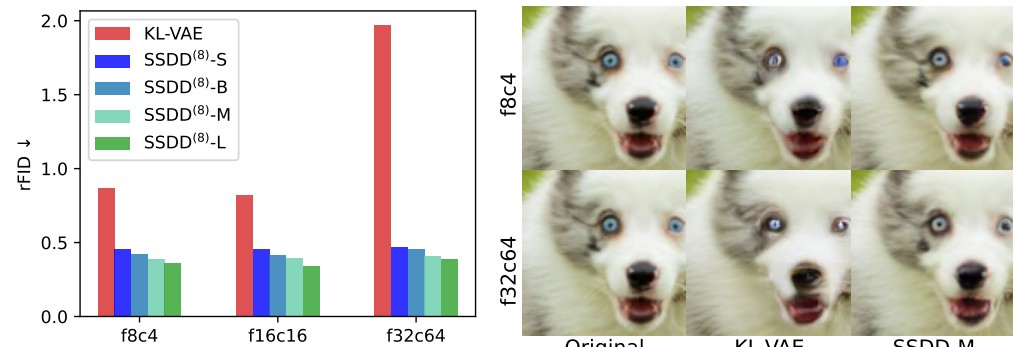

Figure 3: **Evolution of rFID and qualitative comparison when increasing spatial downsampling.** Evaluated on ImageNet $256 \times 256$ with a constant compression ratio by adjusting $c$. Images patches are high-frequency features extracted from a larger $256 \times 256$ image.

the conditional image distribution modeling task complexity increases. Small models (SSDD-S) lack those generation capabilities, but achieve fast, high-quality reconstruction at lower compression rates. SSDD-M strikes a balance by achieving good reconstruction quality at all scales with a parameter count similar to standard KL-VAE models.

While SSDD improves over prior state-of-the-art autoencoders in terms of rFID, LPIPS and Dream-Sim, it yields somewhat worse results in terms of low-level distortion metrics (SSIM, PSNR). We believe, however, that these metrics are not representative for the perceived reconstruction quality — as illustrated in the qualitative examples in Figure 1, Figure S5, and Figure S6 — and should be de-emphasized for evaluation of generation-oriented decoders.

**Scaling across model and image size.** We scale the input resolution from 128 to 512 and model size from **S** to **H** in Table 2, using the f8c4 encoder configuration. Our SSDD models use single-step distilled decoders fine-tuned at either $128 \times 128$ (evaluations at 128), or $256 \times 256$ (evaluations at 256 and 512). Overall, SSDD yields significantly better results than existing baselines, even with smaller models, in particular at 128 and 512 resolution.

**Impact of spatial downsampling.** We conduct evaluations of reconstruction quality on both SSDD and KL-VAE across encoders with different spatial downsampling rates, but using the same total latent space size. In particular, we consider three settings where $f^2 \times c = 16$. In Figure 3 we show reconstruction quality as a function of the downsampling factor. We also visualize the lowest and highest spatial downsamplings in Figure 3, and in Figures S5 and S6. For SSDD we observe a modest decrease in quality when increasing $f$, whereas KL-VAE suffers from a severe degradation

Table 3: **Evaluation of autoencoders on image generation on ImageNet** $256 \times 256$. Generation speed in images/second includes sampling the DiT-XL/2 model as well as pixel decoding.

| | Autoencoder | No CFG | | With CFG | |
|---|---|---|---|---|---|
| | | gFID↓ | Images/s | gFID↓ | Images/s |
| **f8c4** | KL-VAE Rombach et al. (2022) | 19.52 | 7.05 | 7.53 | 3.54 |
| | DiTo-XL-LPIPS Chen et al. (2025d) | 16.55 | 0.87 | 8.14 | 0.77 |
| | SSDD-S | 15.44 | 7.07 | 7.33 | 3.55 |
| | SSDD-B | 15.07 | 7.05 | 7.18 | 3.54 |
| | SSDD-M | 14.59 | 6.98 | 6.89 | 3.52 |
| | SSDD-L | **14.44** | 6.96 | **6.91** | 3.52 |
| **f16c4** | SSDD-S | 16.36 | 27.55 | 8.74 | 13.98 |
| | SSDD-B | 15.53 | 27.20 | 8.48 | 13.89 |
| | SSDD-M | 13.74 | 26.29 | 7.56 | 13.65 |
| | SSDD-L | **13.32** | 25.95 | **7.46** | 13.55 |
| **f16c16** | KL-VAE Rombach et al. (2022) | 32.29 | 27.79 | 26.62 | 14.04 |
| | SSDD-S | 24.87 | 27.57 | 22.86 | 13.98 |
| | SSDD-B | 24.76 | 27.24 | 22.76 | 13.90 |
| | SSDD-M | 23.99 | 26.34 | 22.00 | 13.66 |
| | SSDD-L | **23.94** | 26.02 | **21.94** | 13.57 |

**3.8×**

Table 4: **Ablation of design choices starting from DiTo as baseline.** We indicate the number $N$ of sampling steps used for each model, directly affecting decoding speed.

| Ablation | rFID↓ | | PSNR↑ | | DreamSim↓ | | N |
|---|---|---|---|---|---|---|---|
| DiTo-S-LPIPS *(#D=48.3M)* | 3.17 | | 23.10 | | 0.107 | | 24 |
| | + SSDD-M decoder *(#D=48.0M)* | 2.01 | *-1.16* | 23.38 | *+0.28* | 0.060 | *-0.048* | 24 |
| | + REPA loss | 1.58 | *-0.44* | 22.82 | *-0.56* | 0.104 | *+0.045* | 24 |
| | + replace z-norm by KL | 1.32 | *-0.26* | 23.02 | *+0.20* | 0.098 | *-0.006* | 24 |
| | + logit-normal sampling | 1.30 | *-0.02* | 23.13 | *+0.11* | 0.101 | *+0.003* | 16 |
| | + $t$-spacing sampler | 1.25 | *-0.05* | 23.43 | *+0.30* | 0.097 | *-0.004* | 8 |
| | + EMA | 1.17 | *-0.08* | 23.38 | *-0.05* | 0.097 | *±0.000* | 8 |
| | + shared encoder (§) | 1.07 | *-0.10* | 23.58 | *+0.20* | 0.090 | *+0.030* | 8 |
| | + shared pre-training (**SSDD**[(8)]**-M**) | **1.06** | *-0.01* | 23.62 | *+0.04* | 0.089 | *-0.001* | 8 |
| | + distillation (**SSDD-M**) | 1.13 | *+0.07* | 23.68 | *+0.06* | **0.088** | *-0.001* | 1 |
| *(§) + GAN loss* | *1.07* | *±0.00* | *23.58* | *±0.00* | *0.089* | *-0.001* | *8* |

in performance for the $f = 32$ setting. This confirms that SSDD architecture functions well with varying spatial downsamplings with limited effect on resulting reconstructions.

**Application on image generation.** We conduct image generation experiments by training DiT-XL models (Peebles & Xie, 2023) on ImageNet $256 \times 256$. In Table 3 we report results in terms of rFID and generation speed. We observe that the increased generative capabilities of the SSDD decoders results in higher sample quality across all setting compared to KL-VAE and DiTo, with and without CFG. While our smallest model SSDD-S outperforms all baselines, larger decoders progressively increase the image quality at all encoder configurations. Additionally, we show that SSDD can significantly increase the throughput for similar generation quality. Using a f16c4 encoder, the image is compressed into a $4 \times$ smaller embedding compared to the f8c4 setting. Using a larger decoder (SSDD-L) with DiT-XL/2, we can generate images $3.8 \times$ faster than with the same DiT based on KL-VAE with an f8c4 encoder, with no loss in quality both with and without CFG.

**Ablation of our decoder design choices.** In Table 4, we evaluate the impact of each of our design choices. We start from a DiTo-S model as a baseline: we reproduce the DiTo architecture from Chen et al. (2025d), and reduce the base channel number to $64$, yielding a 48.3M parameters (similar to SSDD-M) decoder with an f8c4 encoder, which we refer to as DiTo-S. We train this model on $128 \times 128$ images, using the same optimizer and parameters as SSDD. We then progressively add components of our method. Each one improves the generative capabilities of the model (rFID), with some increasing the image-to-image distortion. In particular, our improved architecture and its regularization with the REPA loss bring the most improvements of 1.16 and 0.44 rFID respectively. Using a KL-encoder instead of the layer norm of Chen et al. (2025d) has a positive impact on performance, and allows us to use a standard encoder as commonly used to train generative models (Esser et al., 2024; Peebles & Xie, 2023). It focuses our work around the *decoder* component, without requiring specific properties from the encoder. Additionally, the use of a shared encoder (Section 3.4) and shared pre-training (stage 1 in Section 3.2), which improve training efficiency, do not hurt performance. Finally, only the distillation has a slight but negative impact on rFID, but brings the model from 8-steps to 1-step sampling. To demonstrate that we can achieve optimal results with a GAN-free method, we also evaluate the impact of adding a GAN loss. We follow the parametrization of the *adversarial denoising trajectory matching* from (Zhao et al., 2025a), and use it in combination with the setup shown by §. We do not observe any significant impact on rFID and other metrics. A more detailed analysis of the impact of the GAN is included in Appendix E.

## 5 CONCLUSION

We introduced SSDD, a diffusion autoencoder that leverages flow-matching, perceptual alignment and REPA regularization to reach state-of-the-art reconstruction and high generation quality without relying on adversarial training. Through a lightweight distillation strategy, we showed that multi-step diffusion behavior can be compressed into a single-step decoder, yielding up to $3 \times$ faster decoding than $\varepsilon$-VAE-H at similar rFID. On ImageNet, SSDD consistently outperforms KL-VAE and recent diffusion decoders across reconstruction (rFID, LPIPS, DreamSim) and downstream generation (gFID with DiT), with particularly strong gains under high-compression settings. These results

establish SSDD as the first GAN-free, single-step diffusion decoder, combining speed, stability, and generative fidelity, and providing a scalable foundation for future large-scale generative modeling.

## REPRODUCIBILITY STATEMENT

We ensure reproducibility of our results by: (1) providing extensive details about the architecture and training methodology in Section 3, (2) providing the values for all hyperparameters and architecture configurations in Appendix B, (3) providing the source code used to train and evaluate our models as a supplementary material and (4) upon acceptance of the paper, releasing the weights of our models. We believe that together these ensure the reproducibility of our results, and provide the means to confirm the correctness of our implementation, and evaluate our models on other tasks.

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

# A  THEORETICAL MOTIVATIONS

## A.1  DISTORTION-DISTRIBUTION SHIFT TRADE-OFF

As discussed in Section 1, autoencoders have to navigate along the fundamental trade-off between image-to-image **distortion** $\Delta(x, \hat{x})$ that can be minimized by a deterministic decoder, and **distribution shift** $d(P_x, P_{\hat{x}})$, best optimized using a generative decoder. We show here an illustration of this principle on a synthetic example, followed by theoretical justifications of the previous statement. For a more formal analysis of this trade-off, we refer to Blau & Michaeli (2018).

**Motivation: synthetic example.** Let us illustrate this trade-off with a one dimensional synthetic data example. Let $P_x = \mathcal{U}(-2, 2)$ be the source distribution that takes the form of a uniform distribution over $[-2; 2]$, and $E_d(x) = \text{sign}(x) \in \{-1, 1\}$ the encoder that maps source samples to their sign. For the distortion metric $\Delta(x, \hat{x}) = (x - \hat{x})^2$ the optimal decoder is $D^s(z) = z$. We obtain this result by minimization of $\int_0^2 (D^s(1) - x)^2 dx$ and $\int_{-2}^0 (D^s(-1) - x)^2 dx$, which also holds if $D^s(z)$ is a random variable. Using the Kullback–Leibler divergence as the distribution shift metric $d_{KL}(P_x \| P_{\hat{x}})$, the non-deterministic decoder $D^g(z) \sim \mathcal{U}(z - 1, z + 1)$ achieves an optimal score. Indeed, using Gibbs' inequality, $d_{KL}(P_x \| P_{\hat{x}})$ is 0 if and only if $P_x = P_{\hat{x}}$, which is in particular true by using $D^g$, and can only be achieved by a non-deterministic decoder when using $E$ as the encoder. Additionally, computing the distortion and distribution shift for $D^s$ and $D^g$ gives that $D^s$ has a strictly lower distortion and strictly higher distribution shift than $D^g$. This synthetic example illustrates how generative decoders can help navigate the trade-off between both metric groups.

**Distortion-distribution shift trade-off: supporting claims.** We also introduce the following simple claims to support the intuition behind generative decoding by giving theoretical results for boundary cases:

- **There is always a deterministic decoder minimizing a given distortion metric.** We assume $x$ and the decoder outputs are contained in a closed set $X$. If a non-deterministic decoder $D^g(z)$ minimizes distortion, for any $z$, we have:

$$\int_{x \in E^{-1}(z)} \int_{y \in X} \Delta(x, y) P(D^g(z) = y) dy dx \tag{1}$$

$$\geq \int_{y \in X} \min_{\hat{y}} \left[ \int_{x \in E^{-1}(z)} \Delta(x, \hat{y}) \right] P(D^g(z) = y) dy dx \tag{2}$$

$$= \min_{\hat{y}} \left[ \int_{x \in E^{-1}(z)} \Delta(x, \hat{y}) \right] \tag{3}$$

  which yields a deterministic decoder.

- **There is always a non-deterministic decoder minimizing a given distribution shift metric**. We assume a metric such that $P = Q \implies d(P, Q)$ is minimal. Then we use the generative decoder $D^g(z) = u$ with $u \sim P_x$ a random variable, independent of $z$.

- **With a lossy encoder, only a non-deterministic decoder will perfectly model $P_x$.** This result comes directly from applying information theory. We define, for a random variable $x \in X$, a lossy encoder as $E : x \mapsto z \in Z$ such that $H(E(x)) < H(x)$, with $H$ measuring the entropy. Given a deterministic decoder $D^s$ and $\hat{x} = D^s(z)$, we have $H(\hat{x}) \leq H(z)$ (deterministic functions cannot increase entropy), yielding $H(\hat{x}) < H(x)$. As a result, $P_{\hat{x}} \neq P_x$.

These claims do not constitute a general proof that, for images, currently existing non-deterministic decoders offer a strictly better trade-off along the distortion-distribution shift curve. But instead they offer intuitions and motivations for use of generative decoders.

## A.2  SAMPLING FROM LPIPS-REGULARIZED MODELS

We aim to provide here theoretical justification about the sampling behavior of diffusion decoders trained using an additional LPIPS loss (Zhang et al., 2018), which we also explore experimentally

in Appendix C. We observe that the sampling of these models is highly sensitive to the scheduler and the number of steps, as opposed to flow-matching models trained without the additional LPIPS loss.

In this section, we note $x_0$ the initial image, $t$ the noise level, $x_t = (1-t)x_0 + t\varepsilon$ the noised image with $\varepsilon \sim \mathcal{N}(0;1)$ the random noise, $z = E(x_0)$ the image latent representation, $\nu = x_0 - \varepsilon$ the true velocity, $\hat{\nu} = D(x_t, z)$ the predicted velocity, and $\hat{x}_0 = x_t + t\hat{\nu}$ the predicted 1-step reconstructed image. We use $L$ to denote LPIPS loss, and $\mathcal{L}_\lambda(\hat{\nu}|\nu) = \|\hat{\nu} - \nu\|^2 + \lambda L(\hat{x}_0, x_0)$ the reconstruction loss that mixes the flow-matching loss with the LPIPS loss. Additionally, we use the simplified notation $\nabla L = (\nabla_{\hat{\nu}} L)(\hat{x}_0, x_0) = (\nabla_{\hat{\nu}} L)(x_0 + t(\hat{\nu} - \nu), x_0)$, with $\hat{x}_0 = x_t + t\hat{\nu} = x_0 + t(\hat{\nu} - \nu)$. At every training step, the model parameters are updated by taking the gradient over $\mathcal{L}$. We have:

$$\nabla_{\hat{\nu}}\mathcal{L}_\lambda = 2(\hat{\nu} - \nu) + \lambda t \nabla L \tag{4}$$

$$= 2(\hat{\nu} - (\nu - \frac{\lambda t}{2}\nabla L)) \tag{5}$$

$$= \nabla_{\hat{\nu}}\mathcal{L}_0(\hat{\nu}|\nu - \frac{\lambda t}{2}\nabla L). \tag{6}$$

Therefore, optimizing $\mathcal{L}_\lambda(\hat{\nu}|\nu)$ is equivalent to optimizing the flow-matching-only loss $\mathcal{L}_0(\hat{\nu}|\nu - \frac{\lambda t}{2}\nabla L)$. This loss learns the velocity field represented by $\nu - \frac{\lambda t}{2}\nabla L$, which is no longer straight: $(\nabla_{\hat{\nu}} L)L(\hat{x}_0, x_0)$ is not necessarily aligned with $\nu$, and its multiplicative factor $(\frac{\lambda t}{2})$ increases with the noise level. Additionally, since $\nabla L = (\nabla_{\hat{\nu}} L)L(x_0 + t(\hat{\nu} - \nu), x_0)$, the norm of $\nabla L$ may also increase with $t$ due to its $t$ multiplicative factor, and as the predicted velocity error $(\hat{\nu} - \nu)$ should be larger when conditioned on highly-noised images.

This analysis reveals that LPIPS-regularized Flow-Matching decoders learn a non-straight velocity field that shifts during their training, and for which we do not have a closed form. We note that, while this behavior hinders denoising with a large number of steps, as the usual assumptions are no longer met, it also enables control over the trade-off between different metrics by modulating the generated distribution shape at sampling time through the sampling method, which is not offered by straight velocity fields. Additionally, the optimal number of steps of such models is usually small, providing faster distillation and sampling for non-distilled models. We therefore do not try to alter this behavior in this work, but to take advantage of it for fast sampling in our models.

# B  IMPLEMENTATION DETAILS

**Model configurations.**  For the encoder $E$ we follow the standard convolutional architecture from Rombach et al. (2022). Our SSDD decoder architecture is detailed in Section 3.1. We display in Table S1 the configuration for each of its blocks: the number of channels at each level ($1\times1$, $2\times2$, $4\times4$ and $8\times8$ patches) is the base **Channels** times the **Depth multiplier**. The middle transformer operates with the same number of channels as the deepest ResNet, and we indicate the number of transformer blocks. For each decoder configuration we also report the number of parameters as #D.

Table S1: Architecture configurations of SSDD decoders.

| Model | #D | Channels | Depth multipliers | # Blocks |
|---|---|---|---|---|
| SSDD-S | 13.4M | 48 | $\{1, 2, 3, 3\}$ | 8 |
| SSDD-B | 20.2M | 64 | $\{1, 2, 3, 3\}$ | 10 |
| SSDD-M | 48.0M | 96 | $\{1, 2, 3, 3\}$ | 12 |
| SSDD-L | 85.2M | 96 | $\{1, 2, 4, 4\}$ | 16 |
| SSDD-XL | 153.8M | 128 | $\{1, 2, 4, 4\}$ | 16 |

**Training & evaluation.**  We train our models on ImageNet-1k (Deng et al., 2009) using the RAdamW schedule-free optimizer (Defazio et al., 2024), a weight decay of $0.001$ and no scheduler. We use the following loss coefficients: $\lambda_{\text{LPIPS}} = 0.5$, $\lambda_{\text{REPA}} = 0.25$, $\lambda_{KL} = 10^{-6}$. We maintain an exponential moving average of the weights with decay rate $0.999$, starting from 50k iterations. During the first shared pre-training stage, we train on $128\times128$ crops with a constant learning rate of $3 \cdot 10^{-4}$ for 1M iterations. When jointly training the encoder, we use a learning rate

of $10^{-4}$ for increased stability. During the second stage, we fine-tune the weights for 500k iterations, additionally decreasing the learning rate to $10^{-4}$ at the $256 \times 256$ resolution. We use random resizing of the image in the range 128 to 256 during the first stage training, and resizing at the target resolution during the second stage, to which we add random cropping at training resolution (128 or 256) and horizontal flip augmentations. All training resizing operations use Lanczos interpolation for downsampling, while evaluation uses bilinear interpolation to match standard evaluation settings Zhao et al. (2025a); Chen et al. (2025d). For distillation, we use a 7-steps teacher model, to maintain a balance between reconstruction and generative capabilities (see Appendix C). To ensure sharp details, we distill for 50k iterations. We evaluate the speed of our models on a single H200 GPU, using the torch-compiled model for 1,000 iterations over a batch size of 128.

## C  SAMPLING AND DISTILLATION ANALYSIS

**Impact of sampling steps on reconstruction.** Quality of samples from diffusion models usually improves with a higher number of sampling steps. But as noted by Zhao et al. (2025a), diffusion decoders have an *optimal* number of sampling steps, after which reconstruction quality degrades. In Figure S1 we display this effect on SSDD and DɪTᴏ (Chen et al., 2025d). We observe that additional sampling steps have varying impacts on different metrics. A single step maintains the highest similarity in low-level details between the encoded and reconstructed images as measured by PSNR. Increasing the number of steps deteriorates PSNR but improves the rFID and LPIPS metrics. Using the $t$-spacing scheduler from Sargent et al. (2025) with $\rho = 2$ or $\rho = 4$, perceptual-level distortion measured by LPIPS reaches an optimal value around 6 steps, while distribution-level metric rFID is at its lowest at 8 steps. We also see that this remains consistent across different model scales. As we aim to provide a generative-oriented decoder, this confirms our choice of 8 sampling steps for most of our model. We additionally compare to a linear scheduler ($\rho = 1$), using SSDD-M and a DɪTᴏ-XL model trained with LPIPS. We observe a similar effect of increasing sampling steps happening at a slower pace. This confirms that this effect doesn't come from our specific model or sampling choices but from the perceptual regularization shared by SSDD, $\varepsilon$-VAE and LPIPS-regularized instances of DɪTᴏ. As such, setting the number of sampling steps selects the behavior

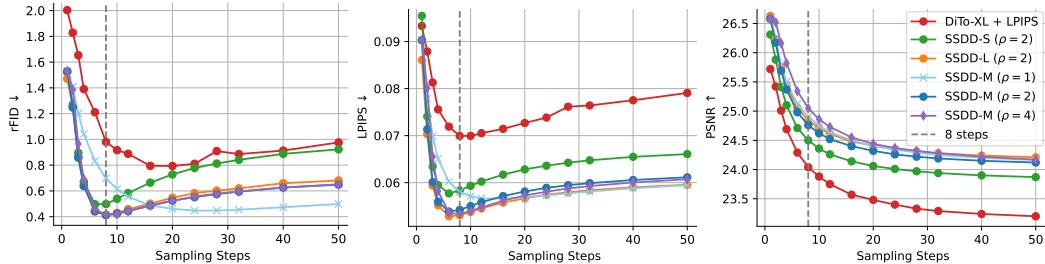

Figure S1: **Evolution of reconstruction metrics depending on the number of sampling steps** $N$**.** Evaluated on ImageNet $256 \times 256$.

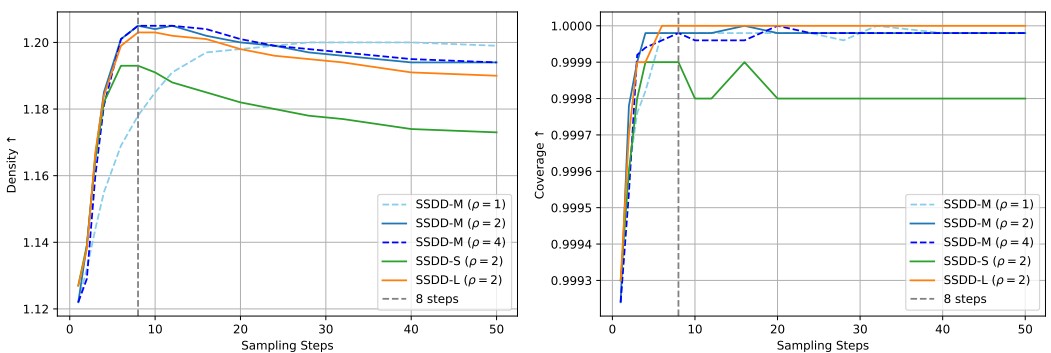

Figure S2: **Effect of sampling on Density and coverage.**

of the decoder relative to the different distortion and distribution shift measures, and distillation consolidates the selected behavior into a single-step model.

**Effect of sampling on fidelity and diversity.** To better understand the behavior behind the shift of model behavior displayed in Figure S1, we measure the Density (fidelity metric) and Coverage (diversity metric) (Naeem et al., 2020) between the ground truth and reconstructed set on ImageNet $256 \times 256$, when varying the number of sampling steps. We evaluate models with diverse sizes (S, M, L) and samplers ($\rho = 0, 2, 4$), and display the results in Figure S2. In each case, both fidelity metrics (FID and Density) drop around the same point. The Coverage stays high even with a high number of sampling steps and does not suffer from meaningful drops. We hypothesize that LPIPS-regularized diffusion decoders suffer from an *overshooting* issue, where the diversity increases outside the bounds of the training distribution support after enough steps.

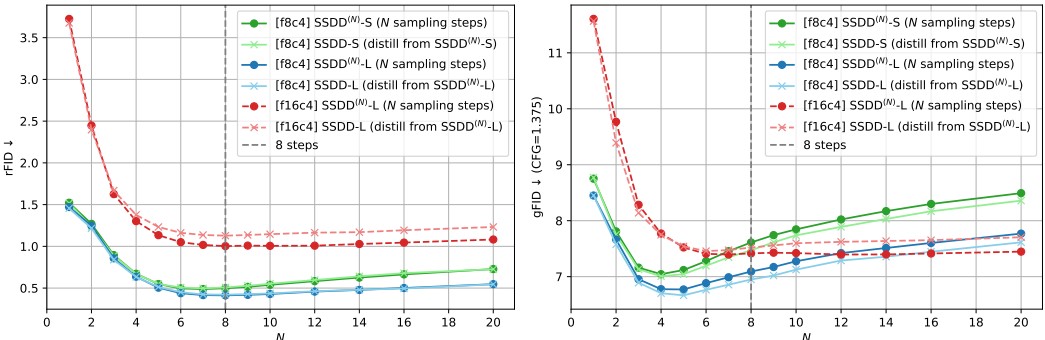

(a) Reconstruction on ImageNet 256x256.

(b) Generation using a DiT-XL/2 trained for 400k steps on ImageNet 256x256, using CFG.

Figure S3: **Effect of the number of sampling steps on reconstruction and generation metrics, evaluated on ImageNet $256 \times 256$.** For distilled models, $N$ is the number of sampling steps of the teacher model. Students are distilled for only $5K$ steps.

**Effect of distillation teacher steps on reconstruction and generation.** We analyze in Figure S3 the behavior of both teacher and distilled student models on reconstruction and generation tasks, depending on the number $N$ of sampling steps. We observe in Figure S3a that the behavior of the distilled models follows closely the teacher on reconstruction tasks when using a low compression rate of f8c4, both for a small (S) and large (L) model. At a higher rate (f16c4), we observe a slight degradation of performances. On the contrary, for generation in Figure S3b the student models display consistent improvements over the teacher models at low compression rates, but at f16c4 a similar (small) performance gap is seen as for reconstruction. This shows that our distillation method preserves most model capabilities for both reconstruction and generative tasks.

Table S2: **Impact of distillation on reconstruction and generation.** The distilled model has a $8\times$ higher throughput for decoding. Results on ImageNet $256 \times 256$, using and 50k distillation steps. Generation results use a DiT-XL/2 trained for 400k steps and a CFG of 1.375.

|  |  | 8-steps sampling | | | Single-step distilled model | | | | | |
|---|---|---|---|---|---|---|---|---|---|---|
|  |  | rFID↓ | DreamSim↓ | gFID↓ | rFID↓ | | DreamSim↓ | | gFID↓ | |
| f8c4 | SSDD-S | 0.50 | 0.046 | 7.61 | 0.50 | ±0.00 | 0.046 | ±0.000 | 6.87 | -0.75 |
|  | SSDD-B | 0.47 | 0.044 | 7.42 | 0.47 | ±0.00 | 0.045 | +0.001 | 6.71 | -0.71 |
|  | SSDD-M | 0.41 | 0.042 | 7.03 | 0.42 | +0.01 | 0.042 | ±0.000 | 6.41 | -0.62 |
|  | SSDD-L | 0.41 | 0.041 | 7.09 | 0.42 | +0.01 | 0.042 | +0.001 | 6.52 | -0.57 |
|  | SSDD-XL | 0.40 | 0.040 | 6.95 | 0.40 | ±0.00 | 0.041 | +0.001 | 6.40 | -0.55 |
|  | SSDD-H | 0.37 | 0.039 | 6.69 | 0.38 | +0.01 | 0.040 | +0.001 | 6.17 | -0.52 |
| f16c4 | SSDD-S | 2.25 | 0.122 | 8.74 | 2.49 | +0.24 | 0.123 | +0.001 | 8.74 | ±0.00 |
|  | SSDD-B | 1.76 | 0.113 | 8.49 | 1.97 | +0.21 | 0.115 | +0.002 | 8.48 | -0.01 |
|  | SSDD-M | 1.16 | 0.099 | 7.55 | 1.31 | +0.15 | 0.101 | +0.002 | 7.56 | +0.01 |
|  | SSDD-L | 0.98 | 0.093 | 7.42 | 1.11 | +0.13 | 0.094 | +0.001 | 7.46 | +0.05 |
|  | SSDD-H | 0.88 | 0.087 | 7.17 | 0.98 | +0.10 | 0.088 | +0.001 | 7.22 | +0.04 |

**Impact of distillation on final model quality.** We control for the impact of single-step distillation on final reconstruction and evaluation results at resolution $256 \times 256$ in Table S2. We follow the distillation procedure from Section 3.3 (details in Appendix B). We evaluate the reconstruction distribution shift (rFID) and perceptual distortion (DreamSim) and the impact on generative models by decoding from a DiT-XL/2. While distillation caused a noticeable impact on the $128 \times 128$ SSDD-M model (see Table 4), we observe slight to no effect at $256 \times 256$ models with an f8c4 encoder, with single-step distilled SSDD even outperforming teacher models on generative tasks. Moving to the higher compression ratio of f16c4, we observe a higher impact on the reconstruction FID, but a reduced impact on generative FID and DreamSim metrics. We conclude that, while distillation can hinder distribution-shift metrics on high compression ratios (f16c4, or f8c4 with smaller images), it has little effect on perceptual quality (DreamSim) and generative applications (gFID), while providing important speed-ups.

# D   ADDITIONAL RESULTS

Table S3: **Reconstruction and generation from frozen pre-trained encoders.** DITO-XL-LPIPS is trained without noise synchronization.

|  | Encoder | Decoder | rFID↓ | LPIPS↓ | DreamSim↓ | PSNR↑ | SSIM↑ |
|---|---|---|---|---|---|---|---|
| f8c4 | SD-VAE (Rombach et al., 2022) | SD-VAE | 0.69 | 0.061 | 0.040 | 25.22 | 0.77 |
|  |  | SD-VAE-XL | 0.64 | 0.059 | **0.035** | **25.37** | **0.79** |
|  |  | SSDD$^{(8)}$-M | 0.47 | 0.060 | 0.046 | 24.44 | 0.75 |
|  |  | SSDD$^{(8)}$-L | 0.45 | 0.058 | 0.044 | 24.52 | 0.76 |
|  |  | SSDD$^{(8)}$-H | **0.41** | **0.056** | 0.043 | 24.67 | 0.76 |
|  | DITO-XL-LPIPS (Chen et al., 2025d) | DITO-XL-LPIPS | 0.78 | 0.102 | 0.058 | 24.10 | 0.71 |
|  |  | SSDD$^{(8)}$-M | **0.52** | **0.062** | **0.049** | **24.64** | **0.76** |

**Reconstruction from existing encoders.** To evaluate the capabilities of SSDD as a replacement for existing decoders, we train our model on top of two existing frozen encoders: SD-VAE (a KL-VAE fine-tuned on a large dataset) and DITO-XL-LPIPS (a diffusion autoencoder). We show in Table S3 that SSDD outperforms the original decoders on reconstruction performance, despite being conditioned on features optimized for a different architecture. This demonstrates that SSDD is a versatile decoder model that can be used to improve the quality from existing auto-encoders.

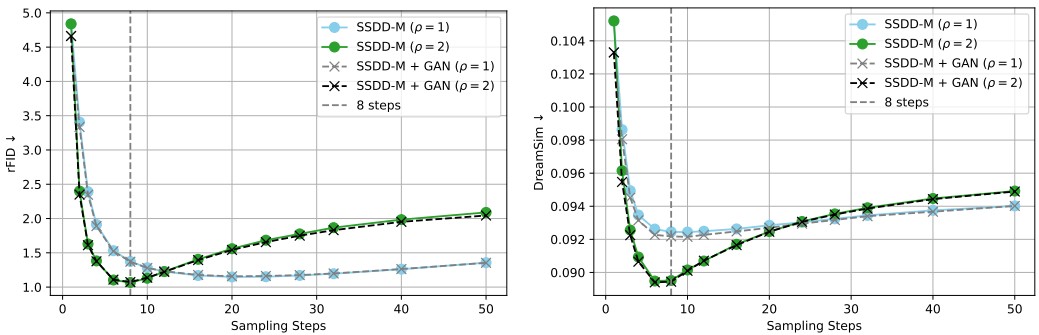

Figure S4: **GAN sampling.** Evolution of the metrics depending on the number of steps $N$. Models directly trained at $128 \times 128$. Evaluated on ImageNet $128 \times 128$.

**Impact of GAN loss on sampling.** According to Zhao et al. (2025a), the fact that a low number of sampling steps is optimal for $\varepsilon$-VAE results from the *denoising trajectory matching*, referring to their adapted GAN loss. This is related to similar work that introduces an adversarial loss on latent diffusion models (Xiao et al., 2022). We test this hypothesis by using the same adversarial loss as Zhao et al. (2025a) on top of our model. We train a single SSDD with an f8c4 encoder at resolution $128 \times 128$, and evaluate with a varying number of steps with linear ($\rho = 0$) and shifted ($\rho = 2$) schedulers. Results are displayed in Figure S4. The adversarial loss has negligible impact

on both the rFID and DreamSim metrics, yielding very slight improvements at some specific points. Additionally, the optimal number of sampling steps for both metrics is not shifted, as the curves keep the same shape. We conclude that the LPIPS loss is the main driver of this behavior, which we also observed in other GAN-free models in Figure S1.

Table S4: **Effect of fine-tuning at target resolution.** We highlight the **first** and second best values of each metric for every evaluation setting.

| Model | Fine-tuned at | Evaluated at $128 \times 128$ | | | Evaluated at $256 \times 256$ | | | Evaluated at $512 \times 512$ | | |
|---|---|---|---|---|---|---|---|---|---|---|
| | | rFID | PSNR | DreamSim | rFID | PSNR | DreamSim | rFID | PSNR | DreamSim |
| SSDD$^{(8)}$-S | N/A (base model) | 2.04 | 23.30 | 0.098 | 0.65 | 24.31 | 0.049 | 0.25 | 26.01 | 0.042 |
| SSDD$^{(8)}$-S | $128 \times 128$ | **1.93** | **23.33** | **0.097** | 0.86 | 24.28 | 0.051 | 0.27 | 25.90 | 0.044 |
| SSDD$^{(8)}$-S | $256 \times 256$ | 2.90 | 23.31 | 0.103 | **0.50** | 24.50 | 0.046 | **0.21** | 26.33 | 0.036 |
| SSDD$^{(8)}$-B | N/A (base model) | 1.64 | 23.39 | 0.095 | 0.51 | 24.59 | 0.046 | 0.31 | 26.42 | 0.037 |
| SSDD$^{(8)}$-B | $128 \times 128$ | **1.54** | 23.44 | **0.094** | 0.55 | 24.60 | 0.047 | 0.35 | 26.35 | 0.039 |
| SSDD$^{(8)}$-B | $256 \times 256$ | 2.22 | 23.27 | 0.099 | **0.47** | 24.60 | 0.044 | **0.22** | 26.56 | 0.034 |
| SSDD$^{(8)}$-M | N/A (base model) | 1.13 | 23.57 | **0.089** | 0.47 | 24.76 | 0.043 | 0.25 | 26.58 | 0.034 |
| SSDD$^{(8)}$-M | $128 \times 128$ | **1.06** | **23.62** | 0.089 | 0.53 | **24.77** | 0.045 | 0.29 | 26.54 | 0.034 |
| SSDD$^{(8)}$-M | $256 \times 256$ | 1.56 | 23.47 | 0.093 | **0.41** | 24.76 | **0.042** | **0.20** | 26.72 | 0.031 |
| SSDD$^{(8)}$-L | N/A (base model) | 0.95 | 23.65 | 0.087 | 0.47 | 24.78 | 0.043 | 0.27 | 26.56 | 0.034 |
| SSDD$^{(8)}$-L | $128 \times 128$ | **0.88** | **23.71** | **0.086** | 0.49 | 24.80 | 0.043 | 0.31 | 26.43 | 0.035 |
| SSDD$^{(8)}$-L | $256 \times 256$ | 1.24 | 23.59 | 0.089 | **0.41** | **24.84** | **0.041** | **0.21** | 26.74 | 0.031 |

**Fine-tuning and generalization at higher resolutions.** To evaluate the impact of target-resolution fine-tuning of our base model trained with multi-resolution data augmentation, we evaluate base and fine-tuned SSDD models at varying resolutions and model sizes, and display the results in Table S4. We observe that for almost all settings and metrics, the base model ranks second and already achieves high-quality results. This ensures that it provides a high-quality shared pre-training for all resolutions with a low training cost.

**Evaluation on COCO images.**

To assess the generalization of our model to other sets of natural images, we follow Zhao et al. (2025a) by evaluating SSDD and comparable baselines on the 5k images from the COCO-2017 Lin et al. (2014) evaluation set. Results are reported in Table S5. We observe that SSDD generalizes better than existing baslines. In particuler, SSDD-B yields better reconstruction than $\varepsilon$-VAE (H), while it was only surpassed by SSDD-L and SSDD-XL on ImageNet in Table 2. This shows that, despite a relatively small size and high decoding speed, our models provide high reconstruction fidelity at various scale and on various data.

Table S5: **Generalization of auto-encoders to new datasets.** All models use an f8c4 encoder and are trained at $256 \times 256$ resolution on ImageNet, except for the consistency decoder. Evaluation is condutec on the COCO-2017 5K validation set.

| Method | #D | COCO $256 \times 256$ | | COCO $512 \times 512$ | |
|---|---|---|---|---|---|
| | | rFID↓ | LPIPS↓ | rFID↓ | LPIPS↓ |
| KL-VAE | 47.2M | 4.65 | 0.063 | 2.68 | 0.064 |
| $\varepsilon$-VAE (M) | 49.33M | 3.98 | - | - | - |
| $\varepsilon$-VAE (H) | 88.98M | 3.65 | - | - | - |
| DiTo-XL + LPIPS | 592.2M | 5.95 | 0.076 | 2.934 | 0.079 |
| Consistency decoder | 625.1M | 4.531 | 0.063 | 2.648 | 0.065 |
| SSDD-S | 13.4M | 3.77 | 0.059 | 2.51 | 0.062 |
| SSDD-B | 20.2M | 3.62 | 0.057 | 2.34 | 0.059 |
| SSDD-M | 48.0M | 3.41 | 0.054 | 2.24 | 0.056 |
| SSDD-L | 85.2M | 3.29 | 0.052 | 2.15 | 0.054 |
| SSDD-XL | 153.8M | **3.25** | **0.050** | **2.13** | **0.053** |

# E  QUALITATIVE RESULTS

We provide a visual comparison of both distilled and non-distilled SSDD with deterministic (KL-VAE) and generative (DITO) decoders in Figures S5 and S6. The same random noise is used to generate each reconstruction for each image, ensuring alignment between distilled and non-distilled model behavior. In Figure S5, using a low spatial downsampling (f8c4), KL-VAE generates distorted details, and DITO outputs blurry low-level features. On the same images, $SSDD^{(8)}$-M generates sharp, low-distortion, and realistic details. Comparing the multi-step and distilled SSDD models, conditioned on the same latent variable $z$ and the same noise $\varepsilon$, their outputs are almost indistinguishable, with the same distortion of low-level features. In Figure S6, using a deeper encoder, the reconstruction quality of KL-VAE drops severely, generating non-realistic images. Our fastest decoder, SSDD-S, still generates similarly sharp details with only slight alterations, while remaining in the range of realistic images. Additionally, larger models (SSDD-M and SSDD-L) output visibly sharper and more accurate reconstructions.

We provide a visualization in Figure S7 of the diversity of images reconstructions by DITO and both multi-step and distilled SSDD. Diversity is not uniformly distributed and is more prominent around details (eyes, numbers, dots) and edges. Both DITO and SSDD diversity maps are focused around the same regions, with smoother and slightly larger areas for the latter. SSDD also benefits from higher diversity levels. The distilled SSDD decoder has similarly diverse generations as the teacher, illustrating how distillation preserves both quality (Figure S5) and diversity (Figure S7).

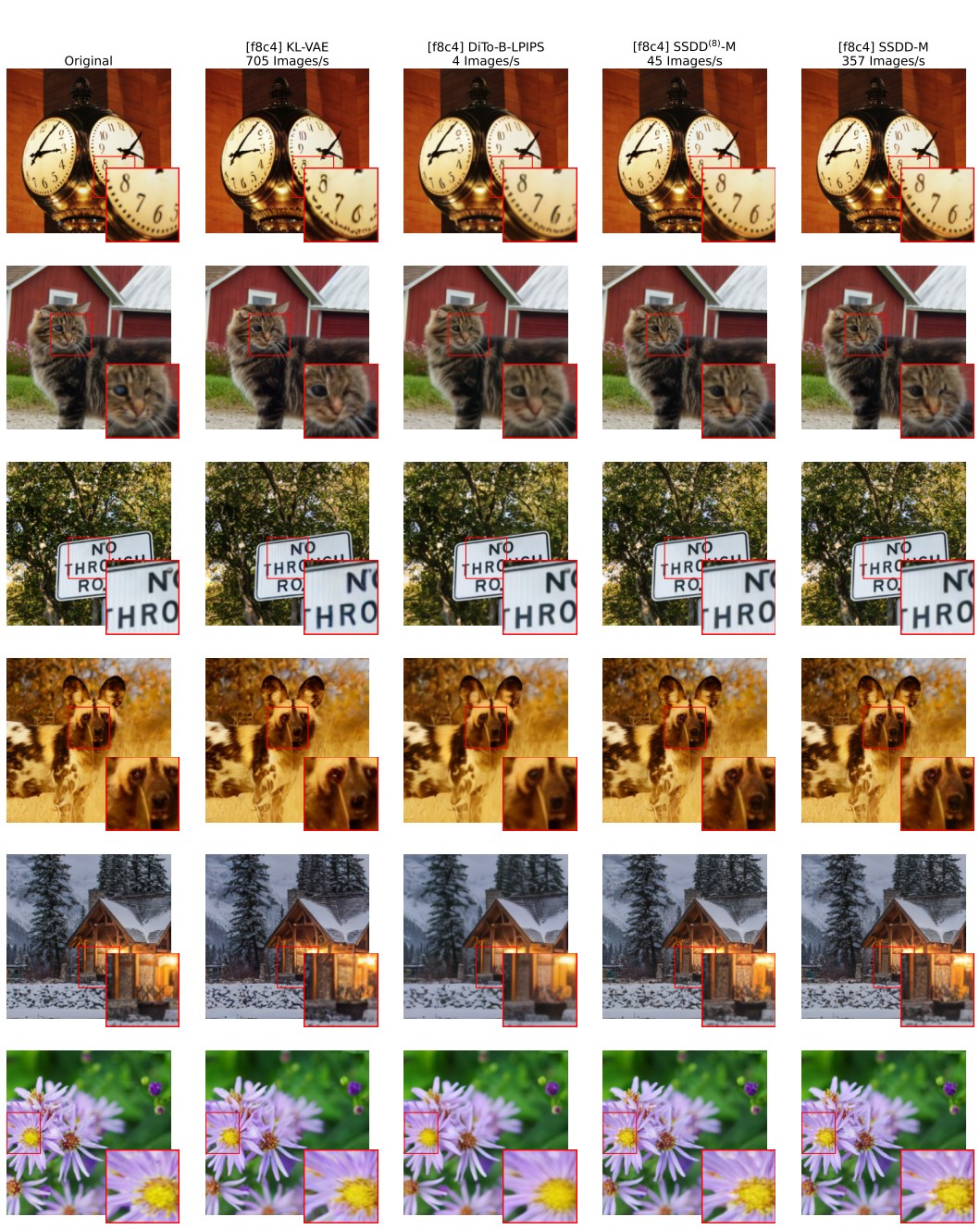

Figure S5: **Qualitative comparison between decoders.** Input images are of resolution $256 \times 256$ and compressed into a $16 \times 16 \times 4$ latent representation by an f8c4 encoder. Left to right: original image, KL-VAE (47.2M parameters), DITO-B (155.2M parameters, 20 sampling steps), SSDD$^{(8)}$-M (48.0M, 8 sampling steps), and SSDD-M (48.0M, distilled into 1 sampling step).

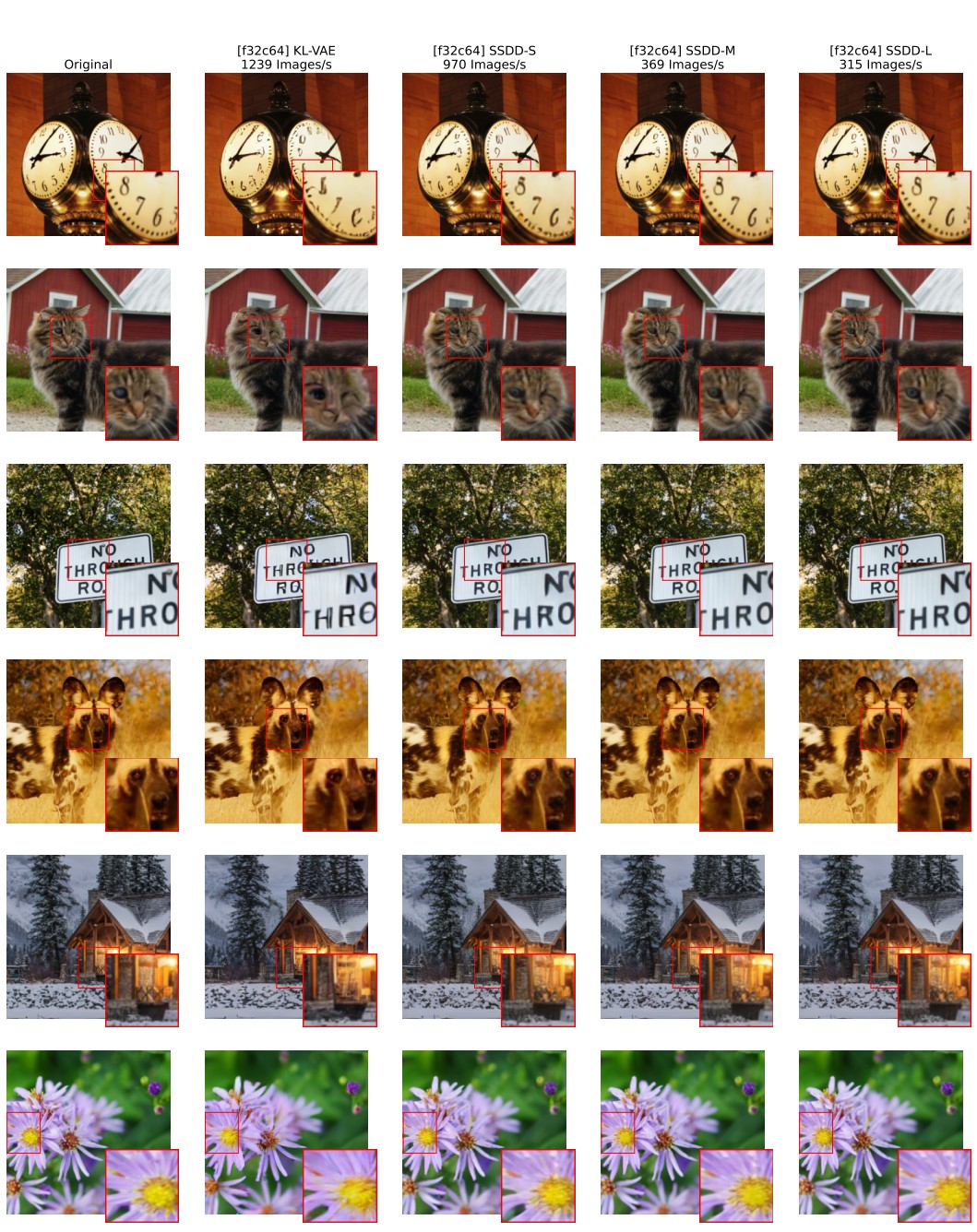

Figure S6: **Qualitative comparison using a deep encoder.** Input images are of resolution $256 \times 256$, and compressed into a $4 \times 4 \times 64$ latent representation by an f32c64 encoder. Left to right: original image, KL-VAE (47.2M parameters), and single-step distilled models SSDD-S (13.4M), SSDD-M (48.0M) and SSDD-L (85.2M).

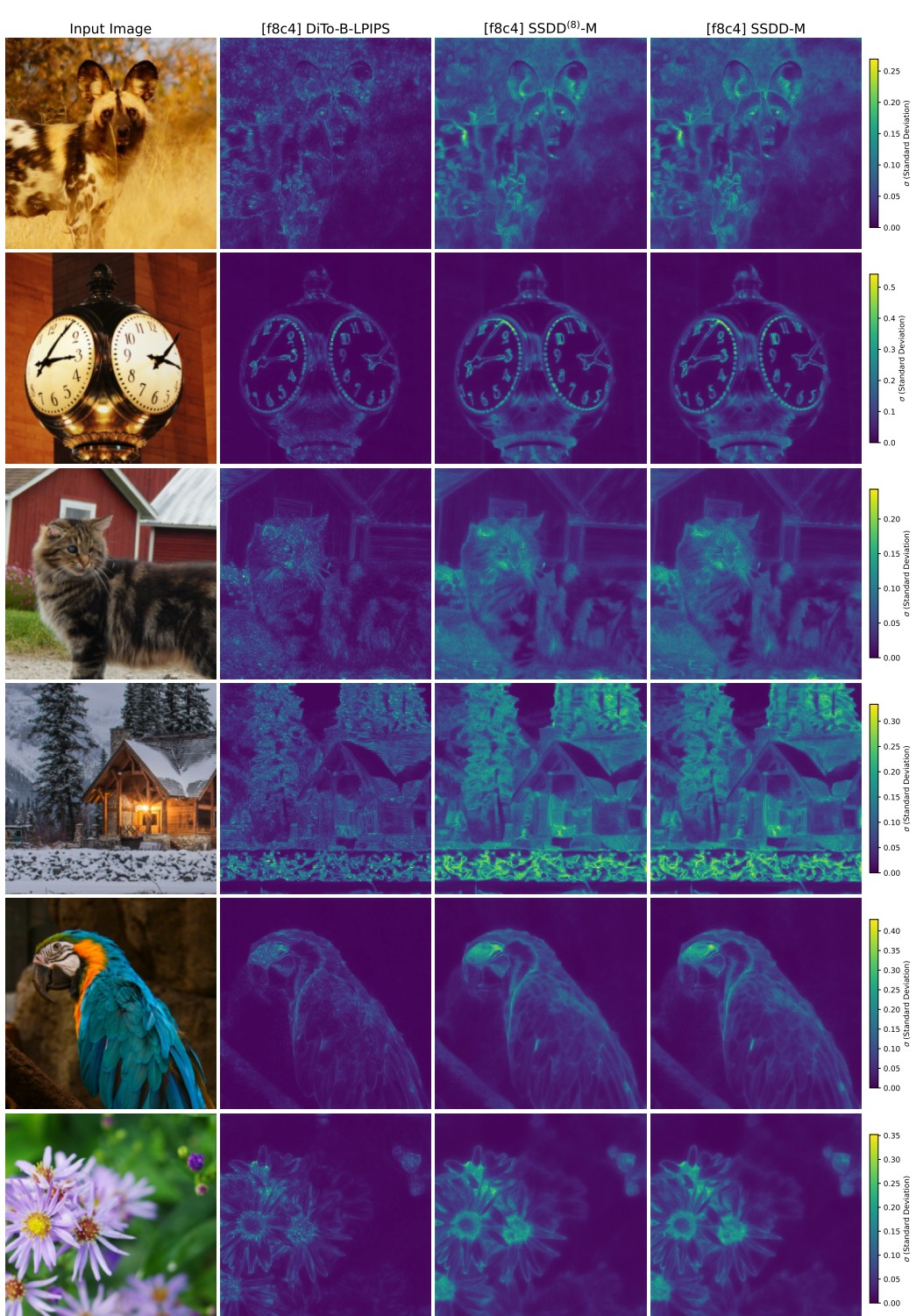

Figure S7: **Diversity maps of reconstructed images.** Left to right: original image, DiTo-B-LPIPS (155.2M parameters, 50 sampling steps), $SSDD^{(8)}$-M (48.0M, 8 sampling steps) and SSDD-M (48.0M, distilled into 1 sampling step). We compute the standard deviation for each pixel over 64 decoding processes for each image. Each row uses a single shared scale to display diversity maps.

