# OpenReview forum: "SSDD: Single-Step Diffusion Decoder for Efficient Image Tokenization"
_ICLR.cc/2026/Conference — Submitted to ICLR 2026_

### Official Review · Reviewer_S7Tn · 2025-10-30

**Soundness:** 3
**Presentation:** 3
**Contribution:** 2
**Rating:** 4
**Confidence:** 3

**Summary:**

This paper proposes SSDD, an efficient single-step diffusion decoder, which achieves improvements in both reconstruction quality and sampling speed without relying on GAN loss.

**Strengths:**

1. The paper introduces SSDD, a single-step decoder, and provides extensive experiments demonstrating its effectiveness in improving reconstruction fidelity and training efficiency.


2. The exploration of removing GAN losses to achieve more stable training is valuable and practically relevant for large-scale diffusion modeling.

**Weaknesses:**

1. This is primarily an engineering-focused work. The proposed SSDD architecture is largely based on U-ViT, with modifications to the resolution hierarchy and the inclusion of REPA loss, LPIPS loss, and a distillation technique. While these design choices lead to strong empirical performance, the methodological novelty and conceptual insights remain limited. Therefore, although this work demonstrates improvements in training efficiency, the insights are relatively limited, which leads me to keep my overall rating at the marginal level.

2. The experiments are conducted mainly on ImageNet. It would strengthen the paper to include additional evaluations on out-of-distribution datasets such as COCO or other domains, to verify the generalization of both reconstruction and generation performance.

**Questions:**

See Weaknesses.

---

> ### Author Response · Authors · 2025-11-20
>
> 1. We would like to thank the reviewer for noticing that these “design choices lead to strong empirical performance”. We address novelty concerns in the general answer. We would like to point out in particular that our paper doesn’t claim theoretical insights, but instead: we show that diffusion decoder can match and outperform KL-VAE decoder both on speed and performance, which was not the case for prior work. This makes SSDD a general and stable-to-train replacement to KL-VAE and other GAN-based decoders.
> Our work can contribute to shift the paradigm used to train autoencoders used by latent generative models, from the common GAN-based approaches to more stable and better performing diffusion-based alternatives.
>
> 2. To verify the generalization of our model, we follow [3] and also evaluate on the COCO dataset to show that SSDD generalize to other sets of natural images. On the COCO 2017 test set, KL-VAE obtains a rFID of 4.65 at 256x256, and 2.68 at 128x128. In comparison, SSDD-B obtains 3.62 and 2.34, respectively. We added the full comparison with multiple baselines and decoder sizes as Table S5 in the revised paper.
>
> [1] Epsilon-VAE: Denoising as Visual Decoding, Zhao et al.
> [2] Diffusion Autoencoders are Scalable Image Tokenizers, Chen et al.
> [3] Scalable Diffusion Models with Transformers, Peebles et al.

---

### Official Review · Reviewer_QJRp · 2025-10-31

**Soundness:** 2
**Presentation:** 2
**Contribution:** 2
**Rating:** 4
**Confidence:** 3

**Summary:**

This paper introduces SSDD (Single-Step Diffusion Decoder), a new image tokenizer designed to replace traditional KL-regularized VAEs. The method's key contributions include: 1) A new pixel diffusion decoder architecture that combines a convolutional U-Net with a central transformer block, designed for improved scalability and stability. 2) A GAN-free training scheme that relies on a combination of a flow-matching objective, a perceptual LPIPS loss, and a REPA feature regularization loss. 3) A distillation process that transfers the high-quality output of a multi-step diffusion decoder into a fast, single-step model. The authors demonstrate that SSDD achieves state-of-the-art reconstruction quality on perceptual metrics like rFID and LPIPS, surpassing existing VAEs and diffusion decoders while offering significantly higher throughput. They also show that using SSDD as a decoder for a DiT model preserves generation quality while drastically speeding up inference.

**Strengths:**

- This paper performs good improvements against baseline.
- Extensive experiments are conducted.

**Weaknesses:**

- The baseline is just $\epsilon$-VAE which is limited.
- What is the difference between train regular decoder and then train a refiner against diffusion decoder?
- Diffusion decoder is more like a generation task instead of reconstruction. The usage of it is questionable.
- There is no novelty in the method side.

**Questions:**

see above

---

> ### Author Response · Authors · 2025-11-20
>
> ### “The baseline is just $\epsilon$-VAE which is limited”.
>
> We would like to point out that we compared against six baselines in the submitted paper. We compare against the best published results of  KL-VAE-based models for reconstruction for each compression setting, which include KL-VAE, SD-VAE, VA-VAE, and LiteVAE (see table 1). Other baselines with underperforming results can be found in referenced papers. For diffusion decoders, we compare both against DiTo and $\epsilon$-VAE. In the revised paper we add comparison to the consistency decoder from DALL-E 3. To the best of our knowledge, these are the only published results for non-discrete diffusion decoder models.
>
> We kindly invite the reviewer to clarify and share any missing baseline that they would like to have seen.
>
> ### “What is the difference between train regular decoder and then train a refiner against diffusion decoder?”
>
> If the reviewer is asking about the difference between regular decoder and our final distilled model: regular decoders are based on a GAN loss, which is prone to instability. The first difference is that our model allows using a GAN-free training method, and our efficient architecture, which outperforms what regular decoders can achieve.
>
> If the reviewer is asking about the difference between the final distillation of our model and training a regular decoder to reproduce the multistep diffusion decoder output: standard decoders learn to map latent encoding $z$ to an image $x$. Our model learns a map from $z$ and pixel-space noise $\varepsilon$ to $x$, increasing the diversity of possible reconstruction. It would be theoretically possible to use a standard decoder to learn that, but has the following limitations: 1) it would need specific architecture changes to condition on pixel-space $\varepsilon$ (while our model keeps the same architecture for distilled and non-distilled). 2) it could require specific tuning to the loss as the model changes, especially if using a GAN loss (while our model keeps the same loss). 3) it would require re-training a new model from scratch and be much more expensive than our distillation process (while for distillation we finetune our model, and are close to convergence in 1/10th of a single epoch). 4) we would have no guarantees that a different model architecture could learn the same map between latent + pixel-noise to pixel-space image, especially as our experiment showed that the model is sensitive to architectural choices.
>
> If the above does not answer the question,  we would like to kindly invite the reviewer to clarify the question.
>
> ### “Diffusion decoder is more like a generation task instead of reconstruction. The usage of it is questionable”
>
> We would like to kindly ask the reviewer to clarify what specifically they find  “questionable”, and in which way the “generation” capabilities of this model would be opposed to “reconstruction”.
>
> As we argue in Section 1 of the paper (2nd paragraph), any reconstruction task from compressed latent with information loss is also a generation task, which matches all the settings we studied. The only difference with diffusion decoders is that they model the distribution of possible reconstruction from a given latent, while deterministic decoder only ever generate just one of those possible reconstructions. Both method families need to infer missing details from the latents.
>
> A model can be considered a decoder if it keeps the same perceptual / semantic information in the images. As can be seen in Table 1, our decoder has a better preservation of these features (LPIPS and DreamSim metrics) than regular decoders such as KL-VAE. This also matches qualitative results from the appendix. As such, we argue that SSDD, despite being trained with a generative loss (as are regular decoders with a GAN loss), is entirely fitted and even better suited for reconstruction. This has already been applied in large-scale generative models, as the consistency decoder from DALL-E 3 uses a similar type of method.
>
> As can be seen in Table 1, our decoder has a better preservation of perceptual / semantic features (LPIPS, DreamSim and FID metrics) than regular decoders such as KL-VAE. This also matches qualitative results in Fig 1, Fig 3, and  last pages of the appendix. As such, we argue that SSDD, despite being trained with a generative loss (as are by the way most regular decoders using a GAN loss), allows for excellent reconstructions. This has already been applied in large-scale generative models, as the consistency decoder from DALL-E 3 uses a similar type of method.
>
> ### “There is no novelty in the method side”
>
> We address novelty concerns in the general answer. We hope the general answer resolves this concern. If not, we would also like to ask the reviewer which of our contributions (as listed in the last paragraph of the introduction) they find not novel and in relation to which prior work.

---

### Official Review · Reviewer_TKd1 · 2025-11-01

**Soundness:** 3
**Presentation:** 3
**Contribution:** 2
**Rating:** 6
**Confidence:** 4

**Summary:**

This paper introduces SSDD (Single-Step Diffusion Decoder), a novel diffusion-based autoencoder for image tokenization that addresses key limitations of existing approaches. The authors propose a hybrid U-Net-transformer architecture that leverages flow matching with perceptual alignment (LPIPS) and REPA regularization to achieve state-of-the-art reconstruction quality without adversarial training. Through a distillation strategy, they compress multi-step diffusion behavior into a single-step decoder, achieving 3.8× faster sampling while maintaining generation quality. SSDD improves reconstruction FID from 0.87 to 0.50 compared to KL-VAE with 1.4× higher throughput, making it suitable as a drop-in replacement for existing tokenizers in generative models.

**Strengths:**

-  SSDD achieves impressive performance improvements across multiple metrics, particularly in reconstruction FID (0.87→0.50) and generation speed (3.8× faster) compared to baselines.
-  Successfully eliminates the need for adversarial training in both encoder training and decoder distillation, while achieving competitive or superior results.
- The paper includes extensive experiments with thorough ablations, multiple baselines, and evaluation across various resolutions (128×128 to 512×512) and model sizes (13.4M to 345.9M parameters).

**Weaknesses:**

- The core architecture builds heavily on existing components (U-ViT, REPA loss, flow matching). While the combination is effective, the individual components are not novel.
- Although SSDD is the first work to demonstrate that a single-step diffusion decoder can match the performance of multi-step diffusion decoders, similar work has already been thoroughly explored in the context of conditional diffusion models (text-to-image or image-to-image). Replacing the condition from text with latents seems not particularly special. This significantly diminishes the contribution of this work.

**Questions:**

See Weaknesses.

---

> ### Author Response · Authors · 2025-11-20
>
> The reviewer’s concerns are mostly about the novelty, which we address in the general answer. We also provide a specific answer to some points raised here.
>
> The reviewer states that “Replacing the condition from text with latents seems not particularly special”. We kindly ask if the reviewer could point to prior work showing that small models conditioned on information dense high-dimensional latents with local per-patch conditioning would perform similarly to text or class conditioned pixel space or latent space diffusion models, as experimental evidence from [2,3] points to the opposite.
>
> We argue that the use of latent-conditioned generative decoder, combined with LPIPS loss for high-quality images, places the model in a very different regime. For instance, the assumption that an increase in sampling step usually leads to increased quality is broken with very few steps (see Appendix C). This breaks the assumption of many distillation techniques that use progressive alignment of the model, where we need to align with a specific selected behavior.
>
> For a more general view, while latent space modeling is becoming a well-mastered task, pixel-space diffusion stays a harder task, with only few models [1] reaching close-to-SOTA results on generation. For the specific task of decoding from latents, previous papers [2,3] have shown that naive application of pixel-space diffusion model on latents is insufficient, and even their improvements were not enough to outperform GAN-based decoder on both reconstruction quality and speed. As such, this application has specific challenges that SSDD tries to overcome. We provide details on how to train such decoder, without relying on a GAN loss, which makes it the first GAN-free and the first diffusion based latent to outperform KL-VAE on both speed and quality.
>
> We hope these additional clarifications resolve the concerns, and would be happy to discuss further.
>
> [1] Simpler Diffusion (SiD2): 1.5 FID on ImageNet512 with pixel-space diffusion, Hoogeboom et al.
> [2] Epsilon-VAE: Denoising as Visual Decoding, Zhao et al.
> [3] Diffusion Autoencoders are Scalable Image Tokenizers, Chen et al.

---

> > ### Comment · Reviewer_TKd1 · 2025-11-25
> >
> > I thank the authors for their reply. I would like to note that the original intent of my review was that "for existing diffusion distillation methods, the conditioning does not affect the methodology itself," rather than "there are no differences in the property between diffusion decoders and T2I diffusion." What the authors need to clarify is "what additional difficulties or challenges exist in distilling diffusion decoders compared to T2I diffusion," rather than "diffusion decoders and T2I diffusion are different." Furthermore, the current state-of-the-art approaches for distilling T2I diffusion are not progressive distillation methods either, such as score distillation methods like DMD2.

---

> > > ### Author Response · Authors · 2025-11-25
> > >
> > > Thanks for clarifying the comment about conditioning.
> > >
> > > First, we would like to point out that our paper focus is around diffusion decoder design and training, and that while we show that they lead to a model that can be distilled into a single-step decoder, exploration of diverse or more advanced distillation methods for diffusion decoders is outside the scope of our work.
> > >
> > > Furthermore, we would like to clarify “what additional difficulties or challenges exist in distilling diffusion decoders”, justifying why we didn’t apply other methods. Methods such as DMD2 [4], while not using progressive distillation, still rely on matching score estimate between teacher model and student for a given $x_t$. As as already been mentionned in Epsilon-VAE [2], and as we show in Appendix C (with theoretical justifications in Apendix A.2), diffusion decoders trained with LPIPS regularization (as is usual for latent to pixel space decoders) exhibit different behaviors depending on scheduling and number of steps due to the effect of both training objectives.
> > >
> > > As such, we parametrize the teacher decoder with a given sampling schedule, which corresponds to a specific distribution of reconstruction given latents. The teacher doesn’t give an estimate of the score function for this specific distribution, as is used in [4]. As such, we cannot directly apply this method, which would align the student with a different behavior than the one selected for the teacher.
> > >
> > > We would like to point out that this issue is not specific to our model (it is also exhibited by models from [2,3]), and would like to ask the reviewer if any additional clarification on this is required.
> > >
> > > [4] Improved Distribution Matching Distillation for Fast Image Synthesis

---

> > > > ### Comment · Reviewer_TKd1 · 2025-11-26
> > > >
> > > > Thanks for the reply. I am willing to keep the positive score.

---

### Official Review · Reviewer_8oNA · 2025-11-08

**Soundness:** 2
**Presentation:** 1
**Contribution:** 2
**Rating:** 4
**Confidence:** 4

**Summary:**

This paper presents SSDD, a single-step diffusion decoder designed for efficient image tokenization. The goal is to overcome the inefficiency of KL-regularized VAEs and multi-step diffusion decoders, which typically require adversarial losses and iterative sampling. SSDD introduces a U-ViT–based pixel diffusion decoder trained under a GAN-free flow-matching objective and perceptual regularization, followed by a distillation stage that compresses a multi-step diffusion process into a single-step model. The method claims to improve reconstruction FID from 0.87 to 0.50 and increase decoding throughput by 1.4×, while maintaining the generation quality of DiT models with 3.8× faster sampling. Experiments on ImageNet demonstrate quantitative advantages over KL-VAE, SD-VAE, LiteVAE, ε-VAE, and VA-VAE, across multiple compression setups.

**Strengths:**

- The work targets a practical and relevant limitation in generative pipelines — the trade-off between reconstruction quality and decoding speed. The single-step distillation from a multi-step diffusion decoder is an intuitive and useful engineering contribution that could simplify downstream diffusion training.

- The model design is computationally efficient (U-ViT backbone, GAN-free objective) and can potentially serve as a drop-in replacement for VAEs in large-scale text-to-image systems.

**Weaknesses:**

**Soundness**

My main concern is with the quantitative evaluation. Unless I’m missing something, several reported numbers—especially in Table 3—look inconsistent with established baselines. For instance, the paper claims substantial gaps for the KL-VAE f8c4 tokenizer on ImageNet 256×256; the no-CFG FIDs in the teens and with-CFG FIDs around 6.x are noticeably worse than what is typically reported for comparable setups. This discrepancy makes it difficult to interpret the claimed SSDD gains. Please clarify:

- the exact evaluation pipeline (data preprocessing, Inception network/version, number of samples, seeds),
- the CFG scale search protocol and whether scores are reported with best-searched CFG,
- whether your KL-VAE/SD-VAE checkpoints and training strictly reproduce the original implementations.*

**Technical contribution**

The “new” elements are largely combinations of existing ideas. The authors are essentially applying standard distillation methods to diffusion models for diffusion decoders, without providing significant insights.

**Presentation**

The presentation of this paper has significant room for improvement. I highly recommend the authors to polish their paper to a high standard of English. Sentences in this paper are often incomplete or ungrammatical. Examples:

- 'Common tokenizers such as the KL-VAE from Rombach et al. (2022) are optimized with L1 reconstruction loss, LPIPS (Zhang et al., 2018), and a GAN discriminator (Goodfellow et al., 2014), to which a KL-regularization of the latent space is added.' What does the 'which' refer to here is unclear. It will not be a major obstacle for an experienced reader to comprehend, but it is not grammatically correct.

- 'Pixel-space diffusion decoders mainly leverage the same convolutional U-Net architectures (Zhao et al., 2025a) that were found successful in early pixel-space diffusion models (Dhariwal & Nichol, 2021).' I believe there is a typo.

**Questions:**

- Baseline Consistency:
The reported rFID improvement over KL-VAE is substantial, but the baseline result (rFID = 0.87) appears considerably weaker than the commonly cited performance of the official Stable Diffusion VAE. Could the authors clarify their training setup—including optimizer, loss weights, and data preprocessing—and verify whether their reproduction matches the official KL-VAE checkpoint performance on ImageNet 256×256? Without this confirmation, the strength of the reported improvement is difficult to assess.

- Perceptual Quality After Distillation:
Does the single-step distillation process lead to any loss of perceptual fidelity or fine-detail degradation (e.g., texture aliasing, local blurring, or over-smoothing)? The qualitative examples shown in Figure 3 (right) are not convincing, since the reference image is already very blurry and lacks high-frequency details. Including sharper examples or zoomed-in patches from higher-detail regions would make the comparison more informative.

- Generalization and Scalability:
Could the authors discuss whether the proposed decoder generalizes to higher-resolution images (e.g., 512×512 or 1024×1024) or to cross-domain datasets (e.g., faces, artworks, medical images) without retraining? If retraining is required, how sensitive is the single-step distillation process to scale and domain shifts?

---

> ### Author Response · Authors · 2025-11-20
>
> ### Soundness
>
> In order to clear concerns about the soundness of our results, let us detail our evaluation protocol. We would also kindly invite the reviewer if they could be more specific when they state “several reported numbers [...] look inconsistent with established baselines”, and what are the “comparable setups” that are referred to, so we can further clarify any concerns.
>
> Our purpose is not to optimize the gFID scores by optimizing the DiT, but rather to compare, for a given standard DiT model, how our model impacts the image reconstruction. As such, we follow a standard evaluation procedure, as has been done by other works (including DiTo and $\epsilon$-VAE), using DiT configurations from [1].
>
> About the specific example given: “the no-CFG FIDs in the teens and with-CFG FIDs around 6.x are noticeably worse than what is typically reported for comparable setup”. The no-CFG FIDs we report are similar to what is reported by [1] for the same setup (DiT-XL/2 trained for 400k steps). For the KL-VAE autoencoder, we report 19.52 without CFG (table 3, row 1), and [1] reports 19.47 without CFG (table 4, row 12 of [1]).
>
> Some mismatch of those numbers comes from inconsistencies in the evaluation setups across papers that use different evaluation pipelines. For instance, $\epsilon$-VAE reports 11.63 FID by training models for 1M steps. We chose to keep the standard setting from [1], using 400k steps for the comparison between multiple trained models (as in table 4 of [1]), which makes model comparison less computationally expensive.
>
> Let us also clarify the following concerns raised by the reviewer:
> - “the exact evaluation pipeline”: as stated in section 4.1, we use the official implementation of DiT (https://github.com/facebookresearch/DiT) without any change to the model or code, retaining their exact data processing and evaluation pipeline. We train models for 400k steps, as this configuration is used to compare most of their models (see Table 4 in Scalable Diffusion Models with Transformers). This also follows the configuration used by DiTo.
> - “the CFG scale search protocol”: as with all other hyperparameters of the DiT, we do not conduct any search over it (which we state in section 4.1). We compare over the standard configuration of DiT, with and without CFG. With CFG, we use a scale of 1.375, as it matches best-reported results of the DiT paper.
> - “whether your KL-VAE/SD-VAE checkpoints and training strictly reproduce the original implementations.”: as written in section 4.1, we use official checkpoints for those models without re-training: KL-VAE is from https://github.com/CompVis/latent-diffusion, and SD-VAE is from https://huggingface.co/stabilityai/sd-vae-ft-mse
>
> ### Technical contribution
>
> Please see the general answer.
>
> ### Presentation
>
> We thank the reviewer for pointing out a few imprecise sentences, which we corrected in the updated version of the paper.

---

> > ### Author Response · Authors · 2025-11-20
> >
> > ### Questions
> >
> > - **Baseline Consistency.**
> > For the KL-VAE results, we directly evaluate the rFID of the official checkpoint from [2]. Reported rFID for f8c4, f16c16 and f32f64 models are respectively 0.90, 0.87 and 2.04 (see table 8 in [2]). Our evaluation pipeline yields slightly different results using the same weights, getting an rFID of 0.87, 0.82 and 2.93 (see table 1). As for the Stable Diffusion VAE, we report its performance in table 1 under “SD-VAE”, but do not use it as a baseline in the rest of the paper as it was fine-tuned on a large dataset than ImageNet.
> > If there is a specific paper that is referred to here “the baseline result appears considerably weaker than the commonly cited performance”, then we would be happy to give a more detailed answer to clarify any concerns.
> > Regarding the training setup for KL-VAE, we are pointing to the Section 4.1, where we specify that we are using the official checkpoints from [2] for the KL-VAE, ensuring that we use the same exact model.
> > - **Perceptual Quality After Distillation**: The examples in Figure 3 indeed look a bit blurry as they are zoomed-in patches extracted from a larger 256x256 image. They correspond to high-frequency details on which reconstruction is harder. We will clarify this in the updated version of the paper. To see comparison between more images both at full resolution and on zoomed-in patches of high-frequency details, we refer to figures S5 and S6 in the appendix.
> > - **Generalization and Scalability**: We already show in Table 2 that our single-step model generalizes to higher resolutions. As pointed in Section 3.2, results for 512x512 are obtained by using the model fine-tuned and distilled for the 256x256 resolution without any further training. We will also add a clarification about that in Table 2. As suggested by the reviewer, we also added evaluation at the 1024x1024 resolution to the revised version of the paper (Table 2). We observe the same gap between KL-VAE performance (rFID of 0.20) and SSDD performance (rFID of 0.13 for the SSDD-B model).
> >     - To further demonstrate the generalization of SSDD, we follow eVAE [3] and also evaluate on the COCO dataset. On the COCO 2017 test set, KL-VAE obtains a rFID of 4.65 at 256x256, and 2.68 at 128x128. In comparison, SSDD-B obtains 3.62 and 2.34, respectively. We added the full comparison with multiple baselines and decoder sizes as Table S5 in the revised paper.
> >
> >
> > [1] Scalable Diffusion Models with Transformers, Peebles et al.
> > [2] High-Resolution Image Synthesis with Latent Diffusion Models, Rombach et al.
> > [3] Epsilon-VAE: Denoising as Visual Decoding, Zhao et al.

---

### Author Response · Authors · 2025-11-20
**General answer to all reviewers**

We would like to thank the reviewers for highlighting important strengths of our paper, including that our work “targets a practical and relevant limitation in generative pipelines”, that our “model design is computationally efficient” and “can potentially serve as a drop-in replacement for VAEs”, that “SSDD achieves impressive performance improvements”, that our paper “provides extensive experiments demonstrating its effectiveness”, and that the “the exploration of removing GAN losses to achieve more stable training is valuable and practically relevant for large-scale diffusion modeling”.

Some concerns have been raised about both specific technical points of the paper, and about the contributions of this work. While we answer specific questions to each reviewer separately, we would like to provide a general answer to highlight the contributions our work provides to the community.
Main takeaway from our paper: similarly as to how diffusion models have been shown to beat GANs for pixel-space generation [1], we show that they can also replace completely GAN-optimized decoders for the decoding stage of the current latent diffusion models [2]. Existing works using diffusion-based decoders [3,4,5] suffered from diverse drawbacks both in image quality and generation speed. Our results show for the first time that diffusion-trained decoders can provide improvements on both fronts in pixel-space, and contribute to shifting the perspective on optimal decoding methods.
Methodological contribution: far from behind a straightforward task, pixel-space decoding of latents has been an ongoing research topic ([3,4,5]) facing issues about difficulties of pixel-space diffusion as well as sampling speed on higher resolution images. We provide a clear method to overcome these issues, analyzing the impact of each component (Tables 4, S2, S3, S4, Figures S1, S3, S4) and impact on speed, quality (Table 1) and scalability to higher resolutions (Table 2, Figure 3). All together, they allow training a new class of efficient pixel-space decoders that was not possible before.
Technical contribution: we are releasing the training code of our models (already in the supplementary material of the submission), and will release the weights of trained decoders at different compression scales. We believe it provides a valuable contribution to the community, as there are no open-source or open-weight diffusion decoders providing improved results over KL-VAE decoders.
Analytical insights: our paper provides theoretical and analytical insights about the behavior of diffusion decoders for sampling and distillation. This behavior, as already mentioned by [3], differs from usual diffusion decoders. We provide extensive experiments with additional theoretical insights, highlighting the cause of the behavior that affects all LPIPS-regularized diffusion-based decoders beyond SSDD.
Some reviews question the novelty of the contribution, as several components have been applied to others tasks, each individually. Our contribution is not about introducing such a new component on top of an existing method (as some, like REPA [6], were), but about a methodology and design choice leading to “strong empirical performance”, as it was highlighted in the reviews. Existing work on diffusion decoders [3,4,5] has shown that the training of diffusion-based decoders is a non-trivial task, and existing methods suffer from multiple shortcomings. We hope our contributions can provide significant advances to this research direction.
Some reviews also refer to SSDD as an “engineering-focused work”. While we consider design choices, our work is focused on methodological aspects. We provide extensive comparison with baselines on multiple aspects, and experiments highlighting the impact of each component. We do not optimize or engineer aspects such as the dataset or the latent diffusion model, using standard dataset (ImageNet) and diffusion model (DiT).

We thank the reviewers for taking the same to provide helpful comments on our work. We updated the paper to include additional proposed evaluations, and are open to further suggestions if some experiments seem unconvincing.

Additionally, we fixed a technical issue in the training code (already fixed in the code submitted with the supplementary material) hindering training, and updated SSDD results accordingly, providing stronger results on reconstruction tasks. We further removed the ‘H’ model size, which was not providing significantly stronger results than the new L and XL models.

[1] Diffusion Models Beat GANs on Image Synthesis, Dhariwal et al.
[2] High-Resolution Image Synthesis with Latent Diffusion Models, Rombach et al.
[3] Epsilon-VAE: Denoising as Visual Decoding, Zhao et al.
[4] Diffusion Autoencoders are Scalable Image Tokenizers, Chen et al.
[5] Improving Image Generation with Better Captions, Betker et al.
[6] Representation Alignment for Generation: Training Diffusion Transformers Is Easier Than You Think

---

> ### Author Response · Authors · 2025-11-28
>
> We are kindly asking the reviewers that did not answer yet if we resolved their concerns with additional experiments, and if not, how we could further clarify any remaining points.

---

### Meta-Review · Area_Chair_ZBjc · 2026-01-08

**Summary:**

This paper develops a new tokenizer for image generation. Generally, the main concerns raised by reviewers are 1) insufficient novelty (SSDD combines existing components without new methodological breakthroughs, engineering-focused). 2) doubts on technical contribution relevance (diffusion decoder for reconstruction, distillation vs traditional frameworks). The authors have conducted more experiments and provided explanations to address the reviewers' concerns.

Overall, I think this paper is a borderline case, I appreciate the authors' effort in answering reviewers' questions with extensive experiments and participation in the rebuttal phase. However, after reading all the reviewers' comments and the paper, I kind of feel that this paper may be a bit lacking in depth and novelty, thus I lean towards recommending rejection.

**Reviewer Concerns:**

During the rebuttal phase, some of the reviewers' concerns are addressed, including adding evaluations for higher resolution settings, clarifying all evaluation details, and adding more baselines. However, the key concerns regarding the novelty and contribution of the methodology proposed in this paper are still not well addressed.

**Reviewer Scores:**

Currently, all reviewers either kept the score or did not reply to the authors' response. My understanding is that although some questions regarding the evaluations have been kind of addressed by the authors, the reviewers may still keep the score, or only part of them would like to marginally increase the score (given that their rating for contributions are "fair").

---

### Decision · Program_Chairs · 2026-01-26

Reject